# Education and public engagement using an active research project: lessons and recipes from the SEA-SEIS North Atlantic Expedition's programme for Irish schools

Sergei Lebedev[1], Raffaele Bonadio[1], Clara Gómez-García[1], Janneke I. de Laat[1], Laura Bérdi[1], Bruna Chagas de Melo[1], Daniel Farrell[2], David Stalling[3], Céline Tirel[4], Louise Collins[1], Sadhbh McCarthy[1], Brendan O'Donoghue[5], Arne Schwenk[6], Mick Smyth[1], Christopher J. Bean[1], and the SEA-SEIS Team[1]

[1]Geophysics Section, Dublin Institute for Advanced Studies, Dublin, D2, Ireland
[2]Coast Monkey, coastmonkey.ie
[3]Dundalk Institute of Technology, Dundalk, Co. Louth, Ireland
[4]Lycée Français d'Irlande, Dublin, D14, Ireland
[5]St Columba's College, Stranorlar, Co. Donegal, Ireland
[6]K.U.M., Umwelt- und Meerestechnik Kiel GmbH, Kiel, 24148, Germany

*Correspondence to*: Sergei Lebedev (sergei@cp.dias.ie)

**Abstract.** An exciting research project, for example with an unusual field component, presents a unique opportunity for Education and Public Engagement (EPE). The adventure aspect of the fieldwork and the drive and creativity of the researchers can combine to produce effective, novel EPE approaches. Engagement with schools, in particular, can have a profound impact, showing the students how science works in practice, encouraging them to study science, and broadening their career perspectives. The project SEA-SEIS (Structure, Evolution and Seismicity of the Irish Offshore, www.sea-seis.ie) kicked off in 2018 with a 3-week expedition on the Research Vessel (RV) Celtic Explorer in the North Atlantic. Secondary and primary school students were invited to participate and help scientists in the research project, which got the students enthusiastically engaged. In a nation-wide competition before the expedition, schools from across Ireland gave names to each of the seismometers. During the expedition, teachers were invited to sign up for live, ship-to-class video link-ups, and 18 of these were conducted. The follow-up survey showed that the engagement was not only exciting but encouraged the students' interest in Science, Technology, Engineering and Mathematics (STEM) and STEM-related careers. With most of the lead presenting scientists on the ship being female, both girls and boys in the classrooms were presented with engaging role models. After the expedition, the programme continued with follow-up, geoscience-themed competitions (a song-and-rap one for secondary and a drawing one for primary schools). Many of the programme's best ideas came from teachers, who were its key co-creators. The activities were developed by a diverse team including scientists and engineers, teachers, a journalist and a sound artist. The programme's success in engaging and inspiring school students illustrates the EPE potential of active research projects. The programme shows how research projects and the researchers working on them are a rich resource for EPE, highlights the importance of an EPE team with diverse backgrounds and expertise, and demonstrates the value of co-creation by the EPE team, teachers and school students. It also provides a template for a multi-faceted EPE programme that school teachers can

use with flexibility, without extra strain on their teaching schedules. The outcomes of an EPE programme coupled with research projects can include both an increase in the students' interest in STEM and STEM careers and an increase in the researchers' interest and proficiency in EPE.

## 1 Introduction

STEM subjects are recognised by a large majority of people in Ireland as essential for the country's prosperity (SFI, 2015). Yet, most people are not comfortable with STEM, perceiving the subjects as too specialized. Careers in STEM and, specifically, Earth science do not appear attractive to most school students, in large part due to the lack of information on and exposure to them (Neenan and Roche, 2016). Among specific sections of the public and school students, women's and girls' interest in STEM and careers in STEM is still relatively low, and students in disadvantaged areas have insufficient resources and role models to motivate their interest in STEM (SFI, 2015).

These barriers to STEM engagement are not unique to Ireland (e.g., Tytler et al., 2008), although in Ireland, in particular, the public interest in science and the pride in the national achievements in science have long been well below those for the arts (Ahlstrom, 2019). The world-class research performed in Ireland today in many areas of science is something the country can be proud of and inspired by, adding to the public interest in STEM in the long term. In a more immediate and direct sense, however, research projects themselves and the researchers who work on them represent a rich resource for Education and Public Engagement (EPE).

Getting involved in an ongoing research project offers an appealing gateway to STEM to both school students and adults. Interactions with researchers reveal them to the public as friendly, "normal" people, enthusiastic about their jobs. These interactions are effective in alleviating the common stereotype of scientists as isolated, aloof and focussed on obscure or highly specialized experiments. The exposure to real researchers can thus increase the students' interest in STEM and in careers in STEM.

In this paper, we present, as a case study, an EPE programme developed around a major research project. Started by researchers, the programme grew to include school teachers as co-creators and expanded to the national scale. We describe the best practice developed in the course of the programme, draw lessons from its development, discuss some general inferences, and aim to identify useful templates and recipes for EPE projects that connect researchers to school students and to the general public.

## 2. The SEA-SEIS research project
### 2.1 Background

About 90% of the Irish territory is offshore, most of it to the west of Ireland. Beneath the ocean, spectacular mountains and deep valleys show elevation variations of up to 3-4 km. There are extinct volcanoes, with remarkable biodiversity on their slopes. Ireland's largest sedimentary basins are also here, as are its greatest natural hazards: undersea slope failures have caused tsunamis in geologically recent past (e.g., Salmanidou et al., 2017; Georgiopoulou et al., 2019).

In the project SEA-SEIS (Structure, Evolution And Seismicity of the Irish offshore), Earth scientists from the Dublin Institute for Advanced Studies (DIAS) are investigating the dynamic processes within the Earth that have shaped the seafloor and caused intraplate volcanic eruptions in the Irish offshore and the broader Northeast Atlantic (Lebedev et al. 2018, Steinberger et al. 2019, Lebedev et al. 2019a, 2019b). In order to investigate the structure and flow of the rock within the Earth's crust and mantle, 18 seismic stations have been installed at the bottom of the North Atlantic Ocean. The ocean-bottom seismometers

were deployed from the RV Celtic Explorer between 17 September and 5 October, 2018, to be retrieved 18-20 months later. The network covers the entire Irish offshore, with a few sensors also in the UK and Iceland's waters (Fig. 1).

This major project represents the first successful attempt to instrument a large area of the Atlantic Ocean with ocean-bottom seismometers. It is particularly significant for Ireland, an island nation with extensive coastlines and a special place for the sea

in its culture and history. In addition to its pioneering science, the project also features state-of-the-art technology: the compact, ocean-bottom seismometers can withstand the enormous pressure at the bottom of the ocean while recording its tiniest vibrations, as small as nanometres in amplitude.

**2.2 The 2018 SEA-SEIS Expedition**
The 2018 SEA-SEIS expedition covered over 5,000 km in 18 days in the North Atlantic Ocean. Its main purpose was to install the 18 ocean-bottom seismic stations. It also offered spectacular EPE opportunities and provided a focus for the EPE programme that we have developed.

Scientists onboard included the Chief Scientist (SL) and seismology (RB, JdL, CGG) and geology (LB) PhD students. The team also included the engineer who had led the development of the ocean-bottom seismometers (AS) and an expert technician from DIAS (MS). It also included a journalist and digital media expert (DF) and a sound artist (DS). The diversity of the team was its key strength. All the members joined forces in the technical and EPE work, which came across clearly to school audiences onshore.


**3 Development of the EPE programme**

The programme was initiated by researchers, motivated by exploratory conversations with school teachers that indicated that it would be useful and in demand. The activities were funded primarily by small amounts from research-project budgets.


Starting with the realisation that the North Atlantic Expedition was simply too good an EPE opportunity to be missed, the programme grew and expanded through contributions from researchers and other members of the team onboard the Celtic Explorer and from teachers at secondary and primary schools. The programme benefited immediately from synergy with established EPE projects. The Marine Institute, which operates the Celtic Explorer and has its own, long-running EPE

programme (Joyce, 2009; Joyce et al. 2018) offered berths on the ship to a journalist (DF) and a sound artist (DS), who then made important contributions to the development of our programme. Some of the teachers of the Irish Seismology in Schools network (Blake et al., 2008) participated in the programme and also made contributions to its development.

We reviewed best practices of previous EPE projects connected to active research and looked for any ideas that could be

applicable to ours. Teacher-scientist collaboration aiming to promote hands-on, inquiry-based science teaching is, generally, an established approach. In seismology, specifically, the Princeton Earth Physics Project (PEPP) installed seismometers in schools across the US for use in teaching and in science projects (Nolet, 1993; Steinberg et al., 2000; Phinney, 2002), and a number of seismology-in-schools programmes operate elsewhere around the world (e.g., Bullen, 1998; Virieux, 2000; Blake et al., 2008; Denton, 2008; Courboulex et al. 2012; Balfour et al., 2014; Zollo et al., 2014; Tataru et al., 2016). Collaborative,

teacher-scientist research projects improve, on the one hand, the scientists' awareness of classroom practices and, on the other hand, the teachers' understanding of scientific research, exposing each group to the other's culture and highlighting the advantages of integrating scientific inquiry into the curriculum (Gosselin et al., 2003). In the joint scientist-teacher-student research projects, the participants particularly enjoy being part of authentic research in which they could take initiative and feel a sense of ownership (Jarrett and Burnley, 2003). Scientist-educator partnerships can produce new teacher resources and

lesson plans, incorporating cutting-edge research (Madden et al., 2007).

Recent trends in Earth-science outreach (e.g., Drake et al., 2014; Tong, 2014) include the use of video projects in the science classroom (Dengg et al., 2014; Wade and Courtney, 2014) and storytelling via diverse media (Barrett et al., 2014; Moloney and Unger, 2014). Hut et al. (2016) reviewed the theory of effective geoscience communication through audio-visual media,

with a particular focus on television, and identified six major themes and challenges: scientist motivation, target audience, narratives and storytelling, jargon and information transfer, relationship between scientists and journalists, and stereotypes of scientists among the general public. Live video has already been used extensively in the practice of EPE coupled with marine research: ship-to-shore video events have been performed for a number of years by the International Ocean Discovery Program (IODP) (Kulhanek et al., 2014; IODP, 2019). We were able to use the idea successfully in our programme, with a specific

focus on ship-to-classroom video link-ups and with our own event templates (Section 3.2).

McAuliffe et al. (2018) reported on the creation of a science book for 7-12 year olds. The book showcased the importance of STEM in today's society and aimed to give the children their first conceptions of STEM career pathways. Importantly, children were co-creators in the content development, character design, and "try at home" activities offered in the book. As a result,
93% of parents of participating children felt that the children became more interested in science than they were before. The Irish research examples used in the book were also found to shift the perception that major scientific discoveries could only take place abroad (McAuliffe et al., 2018).

Further insight into the state of the art in the relevant EPE practice can be gained through direct communication with the
practitioners. Contacting and talking to a lot of people—teachers, researchers, EPE and communications experts—was of key importance in the development of the present programme. Best practical ideas and essential partnerships emerged from some of these discussions.

Shortly before the start of the North Atlantic expedition, a press release on the project was circulated by DIAS through a PR
company (Alice PR). This triggered broad coverage of the project in the national media (https://sea-seis.ie/media), which then helped, to some extent, to attract schools to the programme. However, it was the direct contact with schools and announcements through national networks of teachers and principals that were the most effective.

**3.1 Before the expedition: Seismometer-naming competition**
The seismologists among us have a natural tendency to give their seismic stations names like S01, S02, S03, etc. This is, indeed, how our offshore sites were referred to at the experiment design stage, when we determined the locations for the seismometers.

Then, just over two weeks before the expedition, we announced a secondary-school competition to name our seismometers. Two weeks is a short time for a teacher who may meet their science or geography class only once a week. Having more time for the competition would have been beneficial as more classes would be likely to participate. However, this was not possible as the expedition started shortly after the beginning of the academic year.

The competition was advertised through the email list of the Irish secondary school principals. To kick it off, we proposed the first name, Brian, ourselves. This was after Brian Jacob (Senior Professor of Geophysics at DIAS, 1989-2001), who led the work on the continental nature of the basins west of Ireland, which resulted in the Irish territory increasing by about a factor of ten.

Twenty schools—nineteen across Ireland and one in Italy—participated in the competition. Our appeal for help in an important research project has generated genuine enthusiasm in students and got them engaged with the project. The teachers used this to have discussions on geoscience and marine science, as well as Irish and international Earth scientists and explorers. Among the winning names (https://sea-seis.ie/competitions/naming-competition), Maude, for example, was named after Maude Delap, the Irish marine biologist, Tom—after Thomas Crean, the Irish seaman and Antarctic explorer, and Charles and Harry–after the American seismologists Charles Richter and Harry Hess.

Some teachers used the competition to talk with their students of the sea in the Irish history and culture. Allód was named for the ancient Irish god of the sea, and Gráinne—for Gráinne Ni Mhaille, the "Pirate Queen," the well-known, 16th-century lord of the Ó Máille dynasty in the west of Ireland. Yet another approach was to let the students' imagination roam free, giving us names like Eve (for "eavesdropping on the Earth"), Gill and Loch Ness Mometer.

Most schools proposed multiple names, and some names were proposed by more than once school. The SEA-SEIS researchers at DIAS selected and announced the winning names prior to the start of the expedition. Apart from the merit of the names themselves, an additional consideration was to have as many schools as possible among the winners. With some winning names proposed more than once, all the schools that submitted entries by the deadline were among the winners. The students in participating classes—now well engaged with the project—were particularly interested in how their seismometer would do. Most winning classes then participated in the live video links to the ship during the expedition, and the students were keen to see the videos and photos of the deployment of the seismometer that they had named.

## 3.2 "It was like speaking to Indiana Jones!" - Ship-to-class video link-ups

Live ship-to-shore video links had been performed in international EPE programmes before ours, in particular by the International Ocean Discovery Program (IODP) using its RV JOIDES Resolution's scientific cruises (Kulhanek et al., 2014; IODP, 2019). Some of the teachers we knew in Ireland had participated in these video links and used them to stimulate their students' interest in STEM.

Following IODP, we started the video link-ups with an introduction from the ship that set the stage for a question and answer session with the students in the classroom. In contrast to the IODP video events, which, with rare exceptions, are led by educators (Kulhanek et al., 2014), all our video links were led by PhD students onboard. We also had to develop our own template for live video events. Because the weather during our expedition was mostly stormy—which is often the case in the North Atlantic,—the initially planned live tours of the ship, as performed on RV JOIDES Resolution's scientific cruises (Kulhanek et al., 2014), would have been dangerous for the presenters and had to be abandoned. Instead, we started each session with a brief greeting by a PhD student, acting as the main host on-board, from outside on the deck. This was followed

with an 8-minute, pre-recorded video with introductions to the project and to the team (https://doi.org/10.5446/43586) and short videos and photos of seismometer deployments.

With the Marine Institute's assistance, we had secured, in advance and at an additional cost, a dedicated satellite broadband connection from the ship. Following JOIDES Resolution's EPE programme's example, we used the video-conferencing software Zoom (https://www.zoom.us). During the video links, all non-essential internet activity on the ship was turned off. With all of that, the connection was of surprisingly high quality. Nevertheless, every video event also included our colleagues at DIAS as the third party in the video conference—and the co-hosts at the "DIAS HQ." They would greet the school audience at the beginning of the livestream and broadcast the pre-recorded videos using their reliable broadband connection. They were also ready to step in if the connection from the ship deteriorated, which happened once, towards the end of a video link, when the ship moved out of the area of the satellite's coverage.

After the introduction to the project, the science and the team, the students were invited to ask questions (Fig. 2). Prior to the video link, they were asked by their teachers to think of some. Most questions that were asked related to the life on the ship, to the project (its goals, hypotheses and methods), to Earth science (earthquakes, volcanoes, tsunamis and other natural hazards; plate tectonics and dynamics of the Earth interior), to the equipment, how it works and how it was developed, to what scientists do in their jobs, and to how one becomes a scientist. The Q&A sessions lasted between 20 and 70 minutes, depending on how much time the classes had and on our schedule.

Because of the limited number of video link-ups that could realistically be performed, invitations were sent only to the participants of the seismometer-naming competition (20, over half of them then requesting a slot) and to the members of the Irish Seismology in Schools network (Blake et al., 2008), operated by DIAS (45, a few of them requesting a slot). We were able to schedule and perform a video link with every teacher who asked for one.

In total, 18 video link-ups were carried out. The classes were in schools all around Ireland, and there was one connection to a school in Italy—reported on by a regional TV station (https://youtu.be/1vBFLKV8nG0). In some schools, we talked to a large class or to two classes in the same room, sometimes with students sitting on the floor in the isles. On Tory Island, Co Donegal, there were only five students in the room, but that was 100% of the students in this secondary school—and on this island. For remote schools, the video-link format opens new possibilities and makes it much easier to arrange interactions of school students with STEM practitioners.

Most presenters on the ship and at DIAS were female. This offered opportunities to all-girl classes and girls in co-educational schools to connect to and identify with their own role models among the scientists.

Both female and male students were clearly excited to chat with scientists and engineers on a ship in the middle of the North Atlantic. According to the teachers, their students would then tell the entire school as well as their parents of this experience, which further broadened the reach of the event. "It was like speaking to Indiana Jones!" was how the students of Lycée Français d'Irlande, Dublin, summarized it.

Some teachers used the engagement with researchers to accompany special Earth science projects, conducted prior to the video link (Fig. 3). Other teachers spent only one or two periods on the video link and the discussions before and after it. Given the already packed curricula, this flexibility was useful and appreciated by the teachers. The video links provided a "low-cost, high-impact" activity, inspiring the students and encouraging their interest in STEM and STEM careers but not putting excessive strain on the teachers' schedules.

### 3.3 Drawing competition for primary schools

The SEA-SEIS Drawing Competition for Primary Schools ran from October to December, 2018. It was advertised on the SEA-SEIS website and in InTouch, the Irish National Teachers' Organisation's monthly magazine (InTouch, 2018).

We invited the students to draw one of our friendly, adventurous seismometers. Noting the primary school children's concern for the seismometers (will they be scared at the bottom of the sea, all alone?), we made sure that the competition announcement mentioned that diving deep into the sea was the seismometers' favourite thing to do. The announced evaluation criteria included relevance, artistic merit and originality.

We received nearly 70 entries in total (https://sea-seis.ie/sea-seis-art-18). Most of these came from two schools in different counties in Ireland and two classes in the same school in Italy. The remainder came from a few other schools in Ireland.

Because this was a primary school competition, we awarded a prize to every student who sent us a drawing. The prize was the 2019-2020 SEA-SEIS Calendar, featuring the art by the students (Fig. 4). Every drawing was printed in the calendar, with the ones ranked highest by the SEA-SEIS-researcher jury printed on a full page, and the others—a few per page.

The feedback from the participating teachers and students indicated that the competition was enjoyable and increased the students' interest in STEM (see Section 4). The number of schools who entered, however, was relatively small. This was, in part, because primary schools already have their own art competitions and may feel they are already busy enough with those. In order to increase the participation in the future, it will be useful to communicate to schools more explicitly how the competition can raise the students' interest in STEM, and also mention that every valid entry will win a prize, which we have not done in this case.

**3.4 Song and Rap competition for secondary schools**

The SEA-SEIS Song and Rap Competition for Secondary Schools ran from October to December, 2018. It was advertised on the SEA-SEIS website and through teacher networks. We invited the students to compose and record a song or a rap on a topic related to seismology, the SEA-SEIS Expedition, Earth science or exploration of the interior of the Earth. For information, we directed the students to the project website, to video link-ups with the ship if their class participated, and to further reading. Entries from entire classes or smaller groups of students were accepted. The evaluation criteria included relevance, scientific insight and accuracy, artistic merit and originality.

The competition received excellent entries—creative, imaginative, artistic, and with a variety of original takes on Earth science and seismology at sea (https://sea-seis.ie/sea-seis-rap-18). The competition Grand Prize was shared by two top entries. Runners-up were distinguished by the Jury of SEA-SEIS researchers with Special Mentions. The Grand Prize winning groups received the SEA-SEIS/DIAS branded flash drives (16GB, waterproof to 100 m depth), one for each student in the group and one for their teacher. These were appreciated by all the recipients (Fig. 5). Classes contributing entries that received Special Prizes and Special Mentions received inspirational science books and 4-colour, SEA-SEIS branded pens, also successful as prizes.

One of the teacher authors of this paper (CT) offered the production of a competition entry as a graded Technology project to one of her classes. This was a successful and effective approach, with the production of an entry comprising research on the science, creation of the piece, recording, and preparation of a report.

It was clear from the entries that many students made an effort to research the subject and learn more of the science. Others, however, seemed less interested to learn, and some succeeded in creating impressive entries in spite of that. At one end of the spectrum, one of the winning compositions referred to most SEA-SEIS seismometers by their names, also with a reference to their location and to what they were recording on the seafloor, which showed substantial research on the subject. At the other end, some of the entries showed little understanding and probably no research behind them.

Drawing a line separating "research" from "no research," however, would be difficult. The competition was developed in order to combine art and science, to get the students create science-themed art and to get them more interested in science. The pieces the students composed and recorded were free-form, and there were no correct answers they could insert into in their songs. For this reason, it is not possible to gauge precisely the amount of research students put into Earth-science research in the course of this activity.

By the very nature of the art-science approach, it is not always possible to measure everything. We do draw a lesson from the first edition of this competition, however: in its future editions, it will be useful to steer students more firmly towards learning and towards communicating science and technology in their pieces.

### 3.5 Ethics

The study complied with the Guidance for developing ethical research projects involving children (Department of Children and Youth Affairs, 2012). No personal information on children was collected. No interactions of project participants with children in the participating schools took place, other than the live video link-ups between the researchers and the classrooms, which were conducted by the teachers on the classroom side. The photographs of the children were supplied by the teachers, who confirmed the consent for their use in the online publication. Data collected in the evaluation survey of teachers were undertaken in accordance with good practice. The survey was anonymous by default. Contributors to this study were under no obligation to become the paper's co-authors.

### 4 Evaluation

Formal evaluations were conducted after all ship-to-classroom video link-ups, using a SurveyMonkey online survey. The survey was anonymous by default, but the names of the school and the teacher could be given as an option. All teachers that we contacted right after the video link, on the same day, responded and completed the evaluation form. Of those contacted 3 days after the video-link (once, the ship lost the internet connection over an entire weekend), half did not respond.

Overall, the teachers rated the educational activity at 4.7 out of 5, on average. 86% reported that the video link encouraged the students' interest in science, with 14% reporting "somewhat encouraged," and none reporting "did not encourage." The respondents also reported that the video links triggered the students' curiosity, showed them that science is part of real life, highlighted the importance of collaborating and broadened their career ideas. Getting to know scientists as "open, friendly people" impressed the students. For the classes who had participated in the seismometer-naming competition, the main highlight was, invariably, seeing the deployment of the seismometer they had named.

The evaluation relating to the drawing competition was informal, based on the feedback from the primary school teachers and students themselves. The bulk of the feedback came in the form of 24 thank-you cards from 7-8 year old students from Abbeyleix South National School, Co. Laois, Ireland. It was clear that the children were encouraged to write the cards by their teacher and that the teacher must have mentioned a number of things that could be included in the cards. However, different children opted to include different things in their text, and the phrasing was their own, different in different cards. Nearly all the students wrote that they enjoyed learning about the seismometers and the project, with some mentioning explicitly that

they explained what they had learned to their parents. All were pleased with their prizes, and happy that everybody got a prize. Some wrote that they would like to work with the SEA-SEIS researchers in the future.

335 This evidence would not yield robust statistical inferences (only one class, influenced by their teacher) but, even though the evidence may be regarded as anecdotal, we consider it encouraging and useful. It confirms that young primary school students are curious about and receptive to the general ideas of Earth science research, and that rewarding every participant with a prize is an effective approach in primary school competitions. It also highlights the key role of an actively participating teacher and the importance of a teacher network in order for such competition to reach a broad, national scale.

340

## 5 Discussion

In this section, we focus on the lessons and recipes provided by our EPE programme. We discuss the approaches shown to be 345 particularly important and useful. We also point out what did not work as expected and why, and consider potential next steps towards an expanded, sustainable EPE programme coupled with Earth-science research projects.

### 5.1 Researchers as EPE leaders

Academic researchers are a source of essential STEM expertise. They also possess genuine enthusiasm for science. Their 350 potential capacity for EPE resource development and EPE activities is remarkable: scientists are creative, resourceful and have excellent technical and computer skills. The researchers' motivation to participate in STEM activities, however, is not universally high, for a number of reasons. The incentive structures (assessment criteria) at academic institutions prioritise research, published in top journals of the field, as well as service to the institutions, including administration and teaching (Lam, 2011; Hillier et al., 2019). In informal networks of international researchers, achievements in research are also 355 prioritised. Outreach work can be dismissed by a colleague with a quick "Ah, it's not science." Misguided as this may be, most researchers have heard this, and the opinion of the community of peers affects not only their self-esteem but also their employment and funding opportunities.

For the academics still keen to develop EPE activities, allocating time for this can be difficult, especially if this is unrelated to 360 any of their current research projects. Also, even though most funding agencies encourage EPE, they often do not provide any funds for it in regular research grants.

Researchers thus tend to leave EPE development to outreach specialists and participate in the activities occasionally. They are often used as presenters in pre-designed EPE activities, which gives them opportunities for improving their communication 365 skills (e.g., Illingworth et al., 2019)—often not their greatest strength, to begin with. However, consistently using researchers

for what many of them do not particularly enjoy or excel at does not, obviously, get the best out of them, while pushing some away altogether.

EPE coupled with active research projects can channel the academic researchers' drive and creativity into the development of spectacular, novel EPE programmes. Not all research projects are equally suitable for this, and smaller projects may lack the scale and personnel. A certain proportion of research projects, however, will always present excellent opportunities for the development of effective EPE programmes, led by scientists or by scientists and educators together. Projects with an exciting field component, in particular, easily capture the imagination of school students and engage them, as illustrated by the SEA-SEIS and a number of other EPE programmes (e.g., Kulhanek et al., 2014; IODP, 2019).

## 5.2 Team with diverse backgrounds

Our programme benefited greatly from its integration of practitioners from different disciplines. The team that developed ideas, produced digital content and conducted the EPE activities included Earth scientists, engineers and technicians, secondary school teachers, a journalist and a sound artist. Our teachers (CT and BOD) made key contributions to the development of classroom-activity ideas, from the early stages of the programme planning. Our journalist and media expert (DF) participated in the programme development since before the expedition and produced a popular blog that covered the expedition, also maintaining professionally the project's digital media presence, increasingly recognised as essential in science communication (Drake et al., 2014). Together with our sound artist (DS), they shot and edited onboard the project-introduction and tour-of-the-ship videos. The sound artist, whose primary work using the sounds recorded on the ship will be presented in a month-long show as part of a major international festival, made sure we were all heard during the video links, even outside in strong winds. The engineers and technicians (AS, MS, LC) presented and explained with authority the technology aspect of the project. All the team members onboard the Celtic Explorer presented their perspectives on the project to the school students, conveying the importance of collaboration and the diversity of backgrounds and skills that is required by a major science project.

## 5.3 Students co-creating with scientists

School students were invited to help scientists and have made a real contribution to the project. Their names for the ocean-bottom stations (Fig. 1) have permanently replaced the tentative S01, S02, etc. After the data are collected, these names (abbreviated when required) will remain attached indefinitely to seismograms in international data repositories. In our two art competitions, the participants have produced pieces that are now themselves effective tools for education and public engagement.

We found that inviting students to become co-creators gets them engaged with enthusiasm. They are, then, motivated to learn more on the project, the scientific hypotheses behind it, and what the scientists do in the course of the project. Even though

the students do not perform any of the project's key technical tasks, co-creation does help to get across the excitement and creative nature of scientific research more effectively than a pure exchange of information would. This, in turn, is likely to contribute to increasing the students' interest in STEM and STEM careers.

### 5.4 Multi-faceted, "low-cost, high-gain" programme

We offered the teachers the flexibility of activities that could be fit into just one or two periods, or used in science and technology lessons and projects, or integrated into the Earth science part of the curriculum, according to their needs at the time. The different activities were inter-related but independent (the naming competition prior to the expedition, live video link-ups during the expedition, and the song and drawing competitions after the expedition). Some classes participated in only one of the activities, others—in two or three, with the teachers choosing what was the most suitable for them. The activities were 410 "low-cost" in the sense of the minimum required classroom time commitment. Their impact was "high gain" when compared to the modest amount of the class-time investment required. The gain is in terms of encouraging the students' interest in STEM and STEM careers, which was achieved thanks to the captivating adventure aspect of the project's fieldwork, engagement of students through co-creation with scientists, and direct, live-video communications between students and scientists.

### 5.5 What should be improved and perspectives

Our competitions were, in a sense, experiments. When announcing them, we could not predict the level of participation in either of them or their effectiveness in promoting STEM. The seismometer-naming competition was successful and got the students who participated in it engaged in the project. The follow-up drawing and song-and-rap competitions produced some 420 excellent entries, but the number of participating schools was lower than expected. A proportion of entries to the song-and-rap competition showed little evidence of the students researching either Earth science or the SEA-SEIS project's scientific goals.

While successful as a proof of concept, the competitions also highlighted what was missing: an effective network of teachers. We worked closely with a few teachers and attracted around 30 more from different schools through project announcements. 425 But many other teachers did not respond to invitations to join our EPE activities, possibly not finding them sufficiently compelling or sufficiently informative. Our aim is to help the teachers to get their students more interested in STEM. In order to do this more effectively and develop our EPE programme further, we would need to grow an extensive, national-scale teacher network, offering the teachers continuing professional development, workshops and resources.

430 An expanded, sustainable EPE programme should also offer more activities. Video links can be performed not only from the ship but from the lab and from other fieldwork locations. The expanded programme can have joint activities with multiple research projects and a wider group of researchers associated with them. The programme can also broaden so as to target adult audiences, as well as school students. Generally, more engagement, co-creation, discussion and debate are needed in order to

get people of all ages more interested, involved and comfortable with STEM subjects (SFI, 2015). Using the approaches, lessons and recipes from the present programme, this can be addressed through the work with schools supported by the development of an effective teacher network, through presenting science through arts, and through the use of a pervasive digital platform. Such expansion of the programme would, however, require dedicated funding—being sought at the moment.

**6 Conclusions**

A research project with an exciting field component presents a unique opportunity for broad public engagement. Educational activities with schools, in particular, can have a profound, lasting impact, showing the students how science works, encouraging them to study science, and broadening their career perspectives. Participation in a real research project and co-creation with scientists gets the students enthusiastically engaged.

The EPE programme presented here as a case study comprised live video link-ups between scientists on a ship in the North Atlantic and students in classrooms and 3 school competitions, before and after the expedition. Survey responses from the teachers confirmed that the video links encouraged the students' interest in STEM. Researchers—both experienced and early-career—could see the real impact of the outreach and got involved with enthusiasm and commitment. The outcomes of an educational programme coupled with a research project can thus include both the school students getting more interested in STEM and STEM careers and researchers getting more experienced and proficient in the education and public engagement work.

This case study offers useful lessons and recipes for EPE programmes coupled with active research projects. First of all, it highlights how the research projects and the researchers working on them are a rich resource for EPE. Researchers can be effective EPE leaders, with the EPE programmes channelling their drive and creativity into the development of effective, novel EPE activities.

Secondly, it illustrates the importance of an EPE team with diverse backgrounds and complementary expertise and skill sets. In EPE with primary and secondary schools, the most essential partners are the school teachers and principals. The development of a network of actively engaged teachers is a pre-requisite of successful EPE with schools and should, if possible, be initiated at the earliest stages of the programme. Beyond that, our programme capitalized on contributions of not only scientists and teachers, but also engineers and technicians and a sound artist. Effective national media campaign around the start of the SEA-SEIS Expedition and EPE programme was orchestrated by our communication manager, working with a partner PR firm. Our digital media presence was maintained primarily by the team's digital journalist, his expertise increasing its effectiveness greatly.

Thirdly, the programme demonstrates the value of co-creation by the EPE team, teachers and school students. Close collaboration with the teachers was essential in planning and developing the programme activities. Getting the school students co-creating with scientists in the course of the school competitions got them engaged and genuinely interested.


Finally, our project can be seen as a template for a multi-faceted, "low-cost, high-gain" EPE programme. Recognising that the school curricular are already packed, so that teachers find it difficult to allocate a lot of time to extra material, we offered them the flexibility of activities that could be fit into just one or two periods or integrated into the Earth science part of the curriculum, according to their needs at the time. We offered a series of different, inter-related but independent activities (the naming competition prior to the expedition, live video events during the expedition, and the song and drawing competitions after the expedition). Some classes participated in only one of the activities, others—in two or three, with the teachers choosing what was the most suitable for them. In the sense of the minimum required classroom time commitment, the activities were low-cost. The high gain, relative to the amount of class-time investment and in terms of increasing the students' interest in STEM and STEM careers, is achieved thanks to the captivating adventure aspect of the project's fieldwork, engagement of students through co-creation with scientists, and direct, live-video communications between students and scientists.


**Video supplements**

In our three video supplements, we present

1) an 8-minute introductory video created for our ship-to-classroom video link-ups (https://doi.org/10.5446/43586);
2) a light-hearted but informative account of instrument deployments in rough weather—an example of the presentation of an aspect of the project to a broad audience (https://youtu.be/i2lmBIpcgfI);
3) an entertaining compilation of selected images and sounds from our competitions for the primary and secondary schools (https://doi.org/10.5446/43587).


**Data availability**

The evaluation survey data are provided in the Supplement.

**Author contribution**

SL prepared the manuscript with contributions from all co-authors. RB, DF and DS produced the project EPE videos. SL, BOD, CT, DF, SM, RB, CGG, JdL and LB developed and ran school competitions. SL and DF performed programme-evaluation surveying. SL, LB, RB, JdL, CGG, AS, DF, DS, LC and BC developed, managed and performed live video link-ups between the ship, classrooms and DIAS. CT and BOD contributed key ideas and co-created the competition and video-

link activities.

**Competing interests**

The authors declare that they have no conflict of interest.

**Acknowledgements**

We thank the Secondary and Primary School Teachers and Principals in Ireland and Italy who participated in SEA-SEIS competitions and video link-ups (www.sea-seis.ie/competitions, www.sea-seis.ie/ship-to-classroom-live-video-link-ups) and without whose dedication and hard work the activities could not be successful. Special thanks to Ruth Wallace, Primary School Teacher at the Abbeyleix South National School, Co. Laois. Constructive comments and suggestions of the referees, Anthony

Lelliott and Penny Haworth, have helped us to improve substantially the first version of the manuscript. We are grateful to Dr Eucharia Meehan (Registrar and CEO, DIAS) for continuous support and to Dr Fergus McAuliffe (iCRAG) for a helpful EPE discussion at the planning stages of this programme. We thank Louise Manifold, curator of the AerialSparks project for Galway2020, for initiating the artist residence programme on the RV Celtic Explorer. The ocean-bottom seismometers for SEA-SEIS are provided by iMARL, the Insitu Marine Laboratory for Geosystems Research hosted by DIAS (https://imarl.ie).

The RV Celtic Explorer is run by the Marine Institute (https://www.marine.ie). We are grateful to Captain Denis Rowan, the crew of the RV Celtic Explorer and Aodhán Fitzgerald, Research Vessel Operations Manager, Marine Institute, for their expert assistance in achieving our scientific and EPE objectives. Marc O'Connor and Lukasz Pawlikowski provided superb ICT support on-board. The SEA-SEIS project is co-funded by the Science Foundation Ireland, the Geological Survey of Ireland, and the Marine Institute (grant 16/IA/4598). We acknowledge additional support from Science Foundation Ireland grants

13/CDA/2192 and 13/RC/2092, the latter cofunded under the European Regional Development Fund. The complete SEA-SEIS Team is listed at https://sea-seis.ie/team.

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

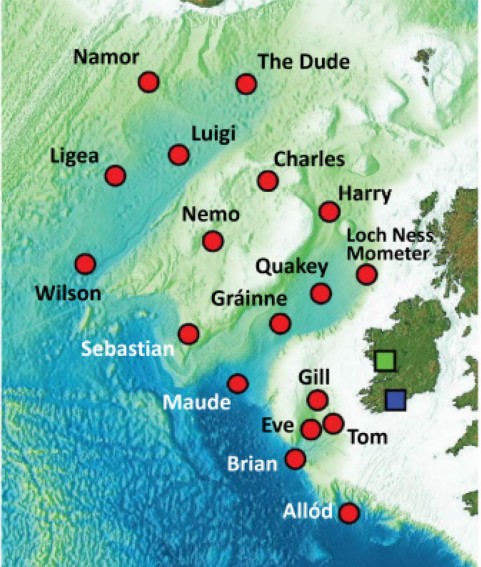
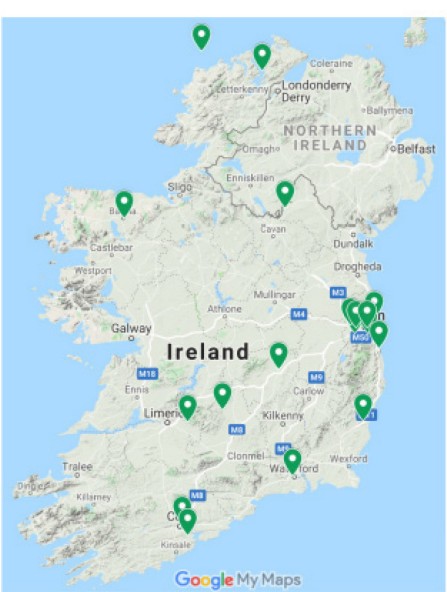

**Figure 1:** Left: the ocean-bottom seismic stations of the SEA-SEIS network (red circles), named by secondary school students in the seismometer-naming competition. Right: the schools across Ireland that suggested the winning names (for an interactive map, see https://sea-seis.ie/competitions/naming-competition).

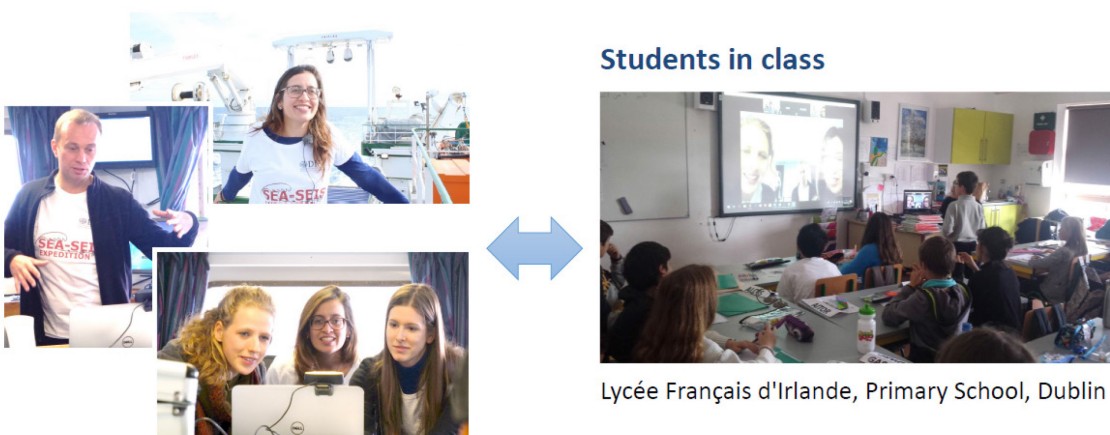

**Figure 2:** Live, ship-to-classroom video link-ups started with a brief introduction of the project and the team and continued with a 20-70 minute Q&A session.

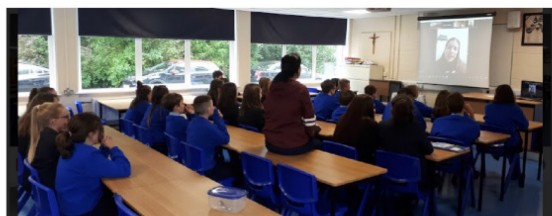
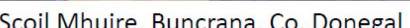
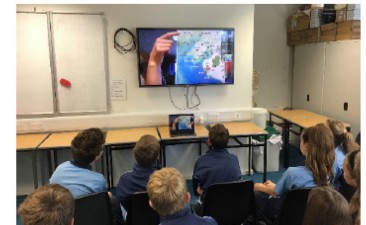
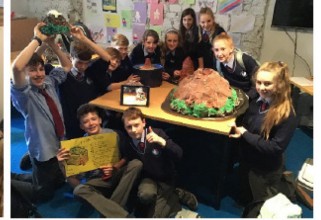

Scoil Mhuire, Buncrana, Co. Donegal

Gaelcholáiste Charraig Uí Leighin, Co. Cork

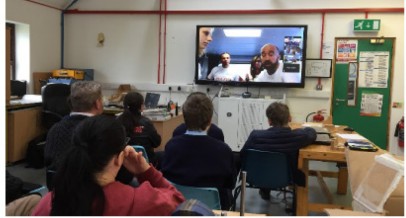
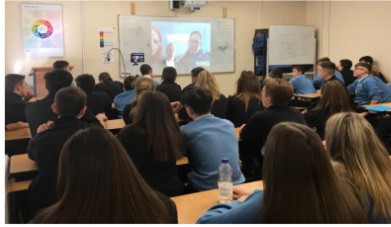
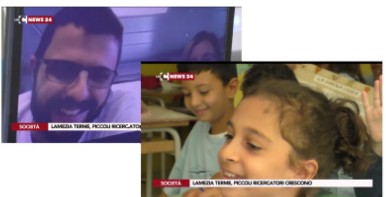

Coláiste Phobail Cholmcille
Tory Island, Co. Donegal

Kingswood Community College, Dublin

I.C. Don Milani Lamezia
Terme, Italy

**Figure 3:** Live, ship-to-classroom video link-ups with different schools in Ireland and Italy.

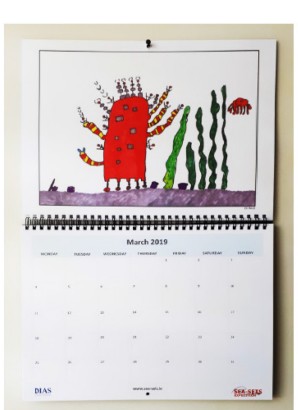
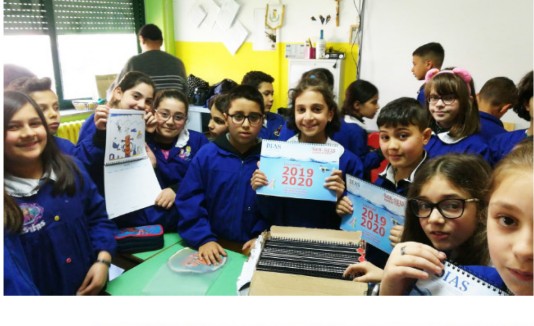
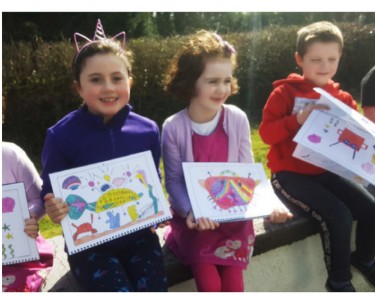
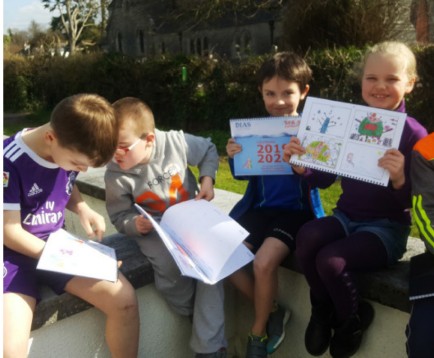

**Figure 4:** Participants of the Primary School Drawing Competition with their prizes, calendars featuring their art. Top left: the 2019-2020 calendar. Top right: students at Istituto Comprensivo Don Lorenzo Milani, Lamezia Terme, Italy. Bottom, left and right: students at Abbeyleix South National School, Abbeyleix, Co. Laois, Ireland.

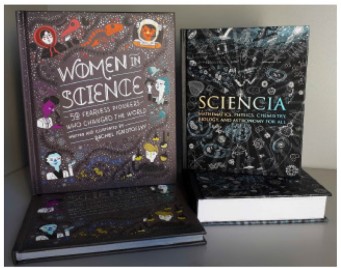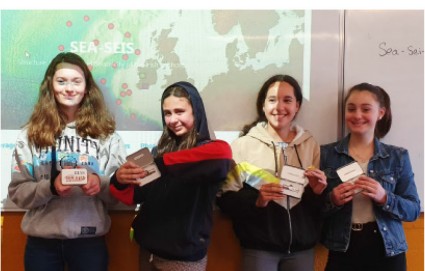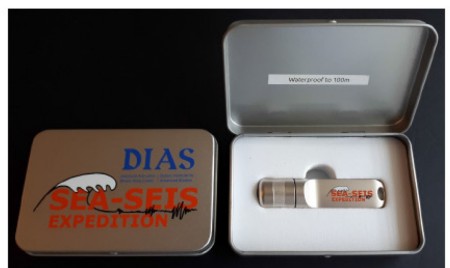

**Figure 5:** Prizes and some of the winners of the geoscience song-and-rap competition for secondary schools. Left:
Inspirational science books went to classes with winning and runner-up groups. Centre: one of the two Grand Prize winning groups (Lycée Français d'Irlande, Dublin). Right: SEA-SEIS branded, 16GB flash drives were awarded to every student in the winning groups and to their teachers.