# Peer review of "Education and public engagement using an active research project: lessons and recipes from the SEA-SEIS North Atlantic Expedition's programme for Irish schools"

_Geoscience Communication, 2019_

## Referee Comment (RC1) · Anthony Lelliott (Referee) · 26 Jul 2019

1. Does the paper address relevant scientific questions within the scope of GC? Yes, the paper is relevant, and clearly relates to Geoscience Communication. 2. Does the paper present novel concepts, ideas, tools, or data? The project itself is novel: scientists working with school students and teachers has been done before but not a project quite like this. The authors need to flesh out the section stating "We reviewed best practices of previous EPE projects connected to active research and looked for any ideas that could be applicable to ours" (p. 4, line 106-7). We need to hear a

summary of such best practices and/or the ideas applicable. 3. Are the scientific methods and assumptions valid and clearly outlined? These are not very clear. The paper is more of a show and tell than providing data about the evaluative aspects of the project. p. 12 line 382: The authors say "No primary data sets were used in producing this article". However, Can they not provide the evaluation data? 4. Are the results sufficient to support the interpretations and conclusions? No, not in several cases. My main issue is that the authors tend to make claims unsupported by any data. For example: p. 6, line 194: Most presenters on the ship and at DIAS were female. This helped all-girl classes and girls in co-educational schools to connect to and identify with their own role models among the scientists. How do you know this? What data is there to support your assertion? p. 7, line 225: The feedback from the participating teachers and students indicated that the competition was enjoyable and increased the students' interest in STEM. Where is the data to support this? How do the teachers actually know that the students' interest was increased? Is this not just the teachers' opinions about a project they liked? p. 11, line 338-9: This contributes to increasing the students' interest in STEM and STEM careers. How do you know this? Line 368: Survey responses from the teachers confirm that the engagement is not only enjoyable but has a lasting positive impact. How do you know this? My point here is that the authors are reading too much into the evaluation data, and making assertions about the positive nature of the experience. They may be right in such assertions, but science communication research needs to be less about describing interesting projects and expecting positive outcomes, and more about interrogating the nature of the projects, and asking difficult questions about them. It would be interesting for the authors to look further into the lack of research on the part of the students: p. 11, line 346-7: "A proportion of entries to the song-and-rap competition showed little evidence of the students researching either Earth science or the SEA-SEIS project's scientific goals" Why was this? What are implications for future EPE projects? 5. Do the authors give proper credit to related work and clearly indicate their own new/original contribution? Yes 6. Does the title clearly reflect the contents of the paper? Yes but

it's too long. I would suggest: "Education and public engagement in a geophysical research project" – or something similar 7. Does the abstract provide a concise and complete summary? Yes, but needs to be edited in line with my comments under (4) above. 8. Is the overall presentation well structured and clear? Yes 9. Is the language fluent and precise? Yes, very well-written 10. Are the number and quality of references appropriate? Yes, but please check the sequencing of some references (alphabetical order).

Minor Points

P. 4 Line 119: "imaginative names" not appropriate (sarcasm?). I would suggest replacing with "unimaginative names".

---

## Editor Comment (EC1) · Marina Joubert (Editor) · 29 Jul 2019

Tony ... you posted a comment 'not sure what this means" ... and now I'm not sure what you refer to. For some reason, I can't seem to find the thread of this discussion. Help please?

---

## Referee Comment (RC2) · Penny Haworth (Referee) · 2 Aug 2019

1. Does the paper address relevant scientific questions within the scope of GC? Yes, the paper is relevant and clearly relates to Geoscience Communication and specifically the topic of Geoscience Education. 2. Does the paper present novel concepts, ideas, tools, or data? I agree with Tony Lelliott's comment that reference to "best practices" and "ideas that could be applicable to ours" (p.4, line 106-7 and line 109-10) need to be fleshed out. Although these are referenced, the authors don't give examples or say why were they chosen or deemed to be applicable. 3. Are the scientific methods

and assumptions valid and clearly outlined? a. The GC scope requires that authors provide qualitative and quantitative evidence – not solely anecdotal reporting and provide quantitative analysis of reach and impact. Whilst the paper is an uplifting account of how the project was implemented, I agree with Tony that the paper is more show and tell and this primarily anecdotal: for example, in line 121, the authors note that "Having more time for the competition would have been beneficial", but do not say why? What had they hoped to achieve? Would having had more time significantly have changed/ improved the results? b. Lelliott indicates that the authors tend to make claims unsupported by any data – see e.g. p.6 line 194 – the assertion is that the girls in all-girl schools identify with female scientists as role models; this assertion is extended to include boys in the abstract (page 1 line 25, "both girls and boys. . .were presented with engaging role models"), and although teachers attested to the positive response of their pupils, the statements are anecdotal rather than substantiated with provable data. In line 199 also page 6, the authors refer to broadened "impact" of the event through pupils talking to their friends and parents about the experience – this certainly can be said to broaden the "reach" of the event, but should not be assumed to be indicative of impact unless that impact can be established and substantiated. Further, at line 225, there is reference to the competition as having "increased the student's interest in STEM" – again there is no data to prove this. c. An example where a conclusion can be linked to the project under discussion to avoid it reading as an unsubstantiated assumption, can be found on page 10, line 332: adding the words "We found that" to the sentence, "Inviting students to become co-creators gets them engaged with enthusiasm" and providing some examples to illustrate the claim, would substantiate the statement which otherwise should be referenced to be considered valid. Further, how do the authors gauge that "They [the students] are, then, motivated to learn more. . ." or that "This contributes to increasing the students' interest in ST and STEM careers" (lines 338-39)? 4. Are the results sufficient to support the interpretations and conclusions? a. Reference to sample size (numbers), where it is made, is imprecise – e.g. Reference is made in the abstract to "18 link-ups", but in line

126 the authors state that "Around 20 schools... participated" – is this 18 -19 schools, 21 schools? The authors need to be consistent in stating the size of the sample set. Should it be the intention to use this paper as a benchmarking exercise for future engagements of this sort, it will be difficult to interrogate data and draw meaningful comparisons. I suggest they indicate exactly how many schools, how many pupils and how many teachers participated – they have provided a distribution map of the schools involved in the project which is useful, and obviously worked closely with teachers at the schools, so substantiated data should be really available. b. Page 8, lines 251-254: In interpreting the results of the competitions, the authors refer to "many" students having "made an effort to research the subject" – again finite number should be available – this could provide useful comparative data for later competitions. On page 11, line 345-347, "the number of participating schools was lower than expected ", and "a proportion of entries showed little evidence of the students having researched..." – But, how do they confirm this is the case and if so what were the factors behind this? Investigating these questions could provide very valuable insights for geoscience education and communication going forward. c. In 'Conclusions', Page 11, lines 367-68, the authors affirm that "Survey responses from the teachers confirmed that the engagement...has a lasting positive impact": I would contend that a single survey with teachers only cannot be said to do this – a claim for "lasting...impact" needs to be measured over time through at least one follow-up survey conducted sometime after the engagement; firstly with pupils (to ascertain retention of knowledge and impact on their view/ understanding of geosciences) and teachers – with regard to how the engagement has supported or augmented their curriculum-based work in the classroom and influenced (especially in the case of high-school pupils) their perfor­mance and results. As far as the "researchers – both experienced and early career", there are a number of considerations that could be used to generate conclusions regarding the success of the outreach: i. E.g. On page 9, line 299, researchers' communications skills are described as, "often not their greatest strength, to begin with": As individuals, how confident were the researchers on this project about being

involved in geosciences engagement and communication before the project? Had this changed afterwards? Why? What are some of the take-home messages that other researchers undertaking engagement might find useful? ii. What were their expectations before the engagement? Did they feel these had been met? Where were the gaps? The authors answer this to some extent by mentioning that the programme should offer more activities (line 356) and broaden the target audience (lines 358-60) in order to achieve their aim (stated in line 353), but it would have been interesting to know how they are going about this – they mention that funding for expansion is "—being sought at the moment" (line 361), so they must feel that the programme was sufficiently successful to demonstrate the need. Who are the targeted funders? Motivating the need for enhanced and broader STEM awareness could be pitched to a variety of potential funders (government, the private sector, industry) but would need to address not only the educational but also societal and economic benefits. How does "the real impact" of the programme move beyond generating "enthusiasm" among researchers and address broader issues to which they can commit their energies and work with societal actors in order to provide answers? 5. Do the authors give proper credit to related work and clearly indicate their own new/original contribution? a. At line 159, the authors indicate they were "keen to use the available best practice" which is acknowledged and credited. However, the reader is not able to compare their approach with other examples as those examples aren't articulated. Later, on p.12 line 382, they say that "No primary data sets were used in producing this article". Some of the best practice examples may have provided a potential source for comparative data and there are enough statements of "fact", or implied measures of comparison, in the article that could be supported by data. b. Not having explained what, in their view, constitutes best practice earlier in the article (ref line 106-7), it is not clear why the project leaders decided to develop a template of their own – was this because there were no examples/reports of live video events available from the best practice examples they had chosen? This would provide an opportunity for them to discuss the implied gaps in the literature and how and why their approach is new, different and an
innovative departure from established geoscience communication practice. 6. Does the title clearly reflect the contents of the paper? a. To an extent, but I'm not convinced that the content has fully articulated the "lessons and recipes" referred to in the title. b. I agree with Tony Lelliott that the title is too long: Suggestion: "Ship to shore – live video involves Irish schools in an active geophysical research project in the North Atlantic." 7. Does the abstract provide a concise and complete summary? a. Yes, but it should be edited to address some of the comments above: e.g. see ref to "boys and girls" being "presented with engaging role models" b. I suggest some of the chosen adjectives in the abstract should be changed to avoid implied assumptions or bias: e.g. i. 'Profound' impact (line 17) – the content doesn't show that the impact of the project has been 'profound'. ii. Line 20-21 – the relative pronoun 'them' in ". . .which got 'them' enthusiastically engaged" is unclear – who became enthusiastically engaged – the students or the researchers? 8. Is the overall presentation well-structured and clear? Yes. 9. Is the language fluent and precise? a. On the whole the article is well-written. b. Some of the expression/idiom is unfamiliar to me – e.g. line 133 "curriculum-facing discussions" – are these curriculum-'based' or curriculum-'relevant'? If the discussions are 'relevant' to the curriculum they could require some interpretation, critical thinking and application by the students; whereas curriculum-'based' discussion would be more guided and possibly require less application and critical thinking. What did the students learn from these discussions? c. Tone: there are two instances where the 'tone' could be interpreted as indicating a 'bias' on the part of the authors: i. On p.4 line 119, the use of the word "imaginative" in the context denotes a certain implied irony – whilst the intention may be to instil some humour, it falls flat. On the other hand it could be interpreted as a typographical error for "unimaginative". ii. On p. 5 line 136-37, the parenthetical reference to the deployment of "Charles" and "Harry", "inevitably, in the UK waters", whilst attempting humour, could be seen as inappropriate in a scholarly work. It assumes also that a global readership would understand and accept the 'joke'. d. Minor errors: i. Line 370 – insert the word "can" after "research project" for the sentence from line 369 to read "The outcomes of an educational programme coupled

with a research project can include..." ii. Line 378 – "form" should read "from". 10. Are the number and quality of references appropriate? Yes. However, the formatting of the reference list is not reader-friendly and makes the references difficult to tell apart. I suggest that the second and following lines of the references are indented to set the references apart. 11. Additional comments: a. I couldn't find any reference to ethical clearance for the research as required by the scope of GC. As the authors are gathering data from schools, and teachers and pupils are case subjects, the research proposal should have received ethical clearance. Is this an oversight? b. Protection of minors: i. Is it acceptable to use photos of under age children on an online platform?

Please also note the supplement to this comment:
https://www.geosci-commun-discuss.net/gc-2019-13/gc-2019-13-RC2-supplement.pdf

---

## Author Comment (AC1) · 5 Sep 2019

Please see, in the supplement to this comment, a pdf file with our response to all comments by the two reviewers, followed by the revised version of the manuscript with tracked changes and a new Supplement for the manuscript, containing evaluation data.

On behalf of the authors,

Sergei Lebedev

[Figure]

Please also note the supplement to this comment:
https://www.geosci-commun-discuss.net/gc-2019-13/gc-2019-13-AC1-supplement.pdf

---

## Author Response (AR1)

*Anthony Lelliott (Referee)*

*1. Does the paper address relevant scientific questions within the scope of GC?*
*Yes, the paper is relevant, and clearly relates to Geoscience Communication.*

*2. Does the paper present novel concepts, ideas, tools, or data?*
*The project itself is novel: scientists working with school students and teachers has been done before but not a project quite like this.*
*The authors need to flesh out the section stating "We reviewed best practices of previous EPE projects connected to active research and looked for any ideas that could be applicable to ours" (p. 4, line 106-7). We need to hear a summary of such best practices and/or the ideas applicable.*

Following these suggestions, the section has been expanded substantially. The sentence quoted by the reviewer is now followed by this new text:

> "Teacher-scientist collaboration aiming to promote hands-on, inquiry-based science teaching is, generally, an established approach. In seismology, specifically, the Princeton Earth Physics Project (PEPP) installed seismometers in schools across the US for use in teaching and in science projects (Nolet, 1993; Steinberg et al., 2000; Phinney, 2002), and a number of seismology-in-schools programmes operate elsewhere around the world (e.g., Bullen, 1998; Virieux, 2000; Blake et al., 2008; Denton, 2008; Courboulex et al. 2012; Balfour et al., 2014; Zollo et al., 2014; Tataru et al., 2016). Collaborative, teacher-scientist research projects improve, on the one hand, the scientists' awareness of classroom practices and, on the other hand, the teachers' understanding of scientific research, exposing each group to the other's culture and highlighting the advantages of integrating scientific inquiry into the curriculum (Gosselin et al., 2003). In the joint scientistteacher-student research projects, the participants particularly enjoy being part of authentic research in which they could take initiative and feel a sense of ownership (Jarrett and Burnley, 2003). Scientist-educator partnerships can produce new teacher resources and lesson plans, incorporating cutting-edge research (Madden et al., 2007).

Recent trends in Earth-science outreach (e.g., Drake et al., 2014; Tong, 2014) include the use of video projects in the science classroom (Dengg et al., 2014; Wade and Courtney, 2014) and storytelling via diverse media (Barrett et al., 2014; Moloney and Unger, 2014). Hut et al. (2016) reviewed the theory of effective geoscience communication through audio-visual media, in particular television, and identified six major themes and challenges: scientist motivation, target audience, narratives and storytelling, jargon and information transfer, relationship between scientists and journalists, and stereotypes of scientists among the general public. Live video has already been used extensively in the practice of EPE coupled with marine research: ship-to-shore video events have been performed for a number of years by the International Ocean Discovery Program (IODP) (Kulhanek et al., 2014; IODP, 2019). We were able to use the idea successfully in our programme, with a specific focus on ship-to-classroom video link-ups and with our own event templates (Section 3.2).

McAuliffe et al. (2018) reported on the creation of a science book for 7-12 year olds. The book showcased the importance of STEM in today's society and aimed to give the children their first conceptions of STEM career pathways. Importantly, children were co-creators in the content development, character design, and "try at home" activities offered in the book. As a result, 93% of parents of participating children felt that the children became more interested in science than they were before. The Irish research examples used in the book were also found to shift the perception that major scientific discoveries could only take place abroad (McAuliffe et al., 2018).

Further insight into the state of the art in the relevant EPE practice can be gained through direct communication with the practitioners. Contacting and talking to a lot of people—teachers, researchers, EPE and communications experts—was of key importance for the present programme. Best practical ideas and essential partnerships emerged from some of these discussions."

*3. Are the scientific methods and assumptions valid and clearly outlined?*

*These are not very clear. The paper is more of a show and tell than providing data about the evaluative aspects of the project. p. 12 line 382: The authors say "No primary data sets were used in producing this article". However, Can they not provide the evaluation data?*

This is a case study, and showing and telling the reader how the programme was developed is an essential part of the paper. The evaluation aspects of the study are, of course, also essential. Following the reviewers' suggestions, we have substantially improved the presentation of the evaluation aspects in the revised version of the manuscript. Evaluation-survey data are now provided in the Supplement.

*4. Are the results sufficient to support the interpretations and conclusions?*

*No, not in several cases. My main issue is that the authors tend to make claims unsupported by any data.*

We accept the criticism and revise the statements pointed out by the reviewers as detailed below, so that they are more precise and do not suggest inferences unsupported by data.

*For example: p. 6, line 194: Most presenters on the ship and at DIAS were female. This helped all-girl classes and girls in co-educational schools to connect to and identify with their own role models among the scientists. How do you know this? What data is there to support your assertion?*

We saw this with our own eyes, the girls discovering that they could have a fun and informative discussion with young female scientists and developing a connection with them. We do not have statistical data for this, however. We thus rephrase the sentence as follows:

> "This offered opportunities to all-girl classes and girls in co-educational schools to connect to and identify with their own role models among the scientists."

The revised statement is weaker but it is certainly accurate.

*p. 7, line 225: The feedback from the participating teachers and students indicated that the competition was enjoyable and increased the students' interest in STEM. Where is the data to support this? How do the teachers actually know that the students' interest was increased? Is this not just the teachers' opinions about a project they liked?*

After this sentence in the text, we have added a reference to Section 4 (Evaluation), where this is explained. In Section 4 (Evaluation) itself, we now explain in greater detail how these inferences were made. The new text on this reads:

> "The evaluation relating to the drawing competition was informal, based on the feedback from the primary school teachers and students themselves. The bulk of the feedback came in the form of 24 thank-you cards from 7-8 year old students from Abbeyleix South National School, Co. Laois, Ireland. It was clear that the children were encouraged to write the cards by their teacher and that the teacher must have mentioned a number of things that could be included in the cards. However, different children opted to include different things in their text, and the phrasing was their own, different in different cards.  Nearly all the students wrote that they enjoyed learning about the seismometers and the project, with some mentioning explicitly that they explained what they had learned to their parents. All were pleased with their prizes, and happy that everybody got a prize. Some wrote that they would like to work with the SEA-SEIS researchers in the future.
>
> This evidence would not yield robust statistical inferences (only one class, influenced by their teacher) but, even though the evidence may be regarded as anecdotal, we consider it encouraging and useful. It confirms that young primary school students are curious about and receptive to the general ideas of Earth science research, and that rewarding every participant with a prize is an effective approach in primary school competitions. It also highlights the key role of an actively participating teacher and the importance of a teacher network in order for such competition to reach a broad, national scale."

*p. 11, line 338-9: This contributes to increasing the students' interest in STEM and STEM careers. How do you know this?*

This became clear on a number of occasions during our ship-to-class video links.  We do not have statistical data on this, however. We thus rephrase the sentence in a weaker form:

"This, in turn, is likely to contribute to increasing the students' interest in STEM and STEM careers."

*Line 368: Survey responses from the teachers confirm that the engagement is not only enjoyable but has a lasting positive impact. How do you know this?*

"Lasting impact" here was used correctly in the context of the sentence: whereas the enjoyment of the activity is limited to its duration, its effect (the encouragement of the students' interest in STEM) is more lasting. We can see, however, how this can get misleading if interpreted differently. We thus remove "lasting impact" and rephrase the sentence in the Conclusions as follows:

"Survey responses from the teachers confirmed that the video links encouraged the students' interest in STEM."

The survey responses that support this statement are included in the data now provided in the Supplement.

*My point here is that the authors are reading too much into the evaluation data, and making assertions about the positive nature of the experience. They may be right in such assertions, but science communication research needs to be less about describing interesting projects and expecting positive outcomes, and more about interrogating the nature of the projects, and asking difficult questions about them.*

Agreed. We add the evaluation data in the supplement and revise a number of potentially misleading statements so as to avoid any inferences more far-reaching than the data warrants.

*It would be interesting for the authors to look further into the lack of research on the part of the students: p. 11, line 346-7: "A proportion of entries to the song-and-rap competition showed little evidence of the students researching either Earth science or the SEA-SEIS project's scientific goals" Why was this? What are implications for future EPE projects?*

We now add the following text in order to discuss and clarify this, with a practical inference for the future in the end:

"At one end of the spectrum, one of the winning compositions referred to most SEA-SEIS seismometers by their names, also with a reference to their location and to what they were recording on the seafloor, which showed substantial research on the subject. At the other end, some of the entries showed little understanding and probably no research behind them.

Drawing a line separating "research" from "no research," however, would be difficult. The competition was developed in order to combine art and science, to get the students create science-themed art and to get them more interested in science. The pieces the students composed and recorded were free-form, and there were no correct answers they could insert into in their songs. For this reason, it is not possible to gauge precisely the amount of research students put into Earth-science research in the course of this activity.

By the very nature of the art-science approach, it is not always possible to measure everything. We do draw a lesson from the first edition of this competition, however: in its future editions, it will be useful to steer students more firmly towards learning and towards communicating science and technology in their pieces."

*5. Do the authors give proper credit to related work and clearly indicate their own new/original contribution?*

*Yes*

*6. Does the title clearly reflect the contents of the paper?*

*Yes but it's too long. I would suggest: "Education and public engagement in a geophysical research project" – or something similar.*

We appreciate the suggestion. We did put a lot of thought into our original title, however, and would very much prefer to keep it. The title suggested by the reviewer limits the scope of the work to a geophysical research project, whereas a very similar programme could be carried out, for example, with a marine geology or marine biology research project. Moreover, a part of our motivation for writing this paper came after our talk on the work in an EPE session at the last EGU meeting, when a colleague working in a space research centre asked if we had a text on our programme that we could share, so that she could use some of our approaches in her outreach work. The first part of the title thus has an appropriately broad scope. As this is a case study, however, it is also important to spell out what this programme was specifically. If the title was just "Education and public engagement using an active research project" or similar, it would give the wrong impression that this is a review covering the very broad area in some detail.  Finally, the "lessons and recipes" are appropriate to mention as this is where most of the general usefulness of the paper probably is. For clarity, lessons are recipes are now mentioned and discussed explicitly and specifically throughout the text (Introduction, Discussion, Conclusions).

The title is, indeed, relatively long, but not exceptionally so: a number of titles on the Geoscience Communication Most Downloaded page (https://www.geosci-commun.net/most_downloaded.html), for example, are similar in length or longer.

*7. Does the abstract provide a concise and complete summary? Yes, but needs to be edited in line with my comments under (4) above.*

The two sentences with unsubstantiated inferences have been rewritten. One now states:

> "The follow-up survey showed that the engagement was not only exciting but encouraged the students' interest in Science, Technology, Engineering and Mathematics (STEM) and STEM-related careers."

The encouragement was observed by the teachers, as documented in our survey, which is now given in the Supplement.

"Lasting impact" mentioned in the original version is now removed. The sentence in the abstract has now been re-written as

> "The outcomes of an EPE programme coupled with research projects can include both an increase in the students' interest in STEM and STEM careers and an increase in the researchers' interest and proficiency in EPE."

It is now a weaker but undoubtedly accurate statement.

*8. Is the overall presentation well structured and clear?*

*Yes*

*9. Is the language fluent and precise?*
*Yes, very well-written*

*10. Are the number and quality of references appropriate?*
*Yes, but please check the sequencing of some references (alphabetical order).*
Fixed – thank you.

*Minor Points*
*P. 4 Line 119: "imaginative names" not appropriate (sarcasm?). I would suggest replacing with "unimaginative names".*
This was self-irony. Self-deprecating humour was helpful in the public engagement programme presented here. However, seeing that this phrasing can be distracting rather than helpful in this paper, we have now removed the word "imaginative" and modified the sentence as follows:

> "The seismologists among us have a natural tendency to give their seismic stations names like S01, S02, S03, etc."

**Reviewer 2.**
*In the full review and interactive discussion, the referees and other interested members of the scientific community are asked to take into account all of the following aspects:*
*1. Does the paper address relevant scientific questions within the scope of GC?*
*Yes, the paper is relevant and clearly relates to Geoscience Communication and specifically the topic of Geoscience Education.*

*2. Does the paper present novel concepts, ideas, tools, or data?*
*I agree with Tony Lelliott's comment that reference to "best practices" and "ideas that could be applicable to ours" (p.4, line 106-7 and line 109-10) need to be fleshed out. Although these are referenced, the authors don't give examples or say why were they chosen or deemed to be applicable.*

The section has been expanded substantially, following these suggestions. The text quoted by the reviewer is now followed by this new text:

> "Teacher-scientist collaboration aiming to promote hands-on, inquiry-based science teaching is, generally, an established approach. In seismology, specifically, the Princeton Earth Physics Project (PEPP) installed seismometers in schools across the US for use in teaching and in science projects (Nolet, 1993; Steinberg et al., 2000; Phinney, 2002), and a number of seismology-in-schools programmes operate elsewhere around the world (e.g., Bullen, 1998; Virieux, 2000; Blake et al., 2008; Denton, 2008; Courboulex et al. 2012; Balfour et al., 2014; Zollo et al., 2014; Tataru et al., 2016). Collaborative, teacher-scientist research projects improve, on the one hand, the scientists'

awareness of classroom practices and, on the other hand, the teachers' understanding of scientific research, exposing each group to the other's culture and highlighting the advantages of integrating scientific inquiry into the curriculum (Gosselin et al., 2003). In the joint scientist-teacher-student research projects, the participants particularly enjoy being part of authentic research in which they could take initiative and feel a sense of ownership (Jarrett and Burnley, 2003). Scientist-educator partnerships can produce new teacher resources and lesson plans, incorporating cutting-edge research (Madden et al., 2007).

Recent trends in Earth-science outreach (e.g., Drake et al., 2014; Tong, 2014) include the use of video projects in the science classroom (Dengg et al., 2014; Wade and Courtney, 2014) and storytelling via diverse media (Barrett et al., 2014; Moloney and Unger, 2014). Hut et al. (2016) reviewed the theory of effective geoscience communication through audio-visual media, in particular television, and identified six major themes and challenges: scientist motivation, target audience, narratives and storytelling, jargon and information transfer, relationship between scientists and journalists, and stereotypes of scientists among the general public. Live video has already been used extensively in the practice of EPE coupled with marine research: ship-to-shore video events have been performed for a number of years by the International Ocean Discovery Program (IODP) (Kulhanek et al., 2014; IODP, 2019). We were able to use the idea successfully in our programme, with a specific focus on ship-to-classroom video link-ups and with our own event templates (Section 3.2).

McAuliffe et al. (2018) reported on the creation of a science book for 7-12 year olds. The book showcased the importance of STEM in today's society and aimed to give the children their first conceptions of STEM career pathways. Importantly, children were co-creators in the content development, character design, and "try at home" activities offered in the book. As a result, 93% of parents of participating children felt that the children became more interested in science than they were before. The Irish research examples used in the book were also found to shift the perception that major scientific discoveries could only take place abroad (McAuliffe et al., 2018).

Further insight into the state of the art in the relevant EPE practice can be gained through direct communication with the practitioners. Contacting and talking to a lot of people—teachers, researchers, EPE and communications experts—was of key importance for the present programme. Best practical ideas and essential partnerships emerged from some of these discussions."

*3. Are the scientific methods and assumptions valid and clearly outlined?*

*a. The GC scope requires that authors provide qualitative and quantitative evidence – not solely anecdotal reporting and provide quantitative analysis of reach and impact. Whilst the paper is an uplifting account of how the project was implemented, I agree with Tony that the paper is more show and tell and this primarily anecdotal: for example, in line 121, the authors note that "Having more time for the competition would have been beneficial", but do not say why? What had they hoped to achieve? Would having had more time significantly have changed/ improved the results?*

To respond first to the specific question in the end of the comment, having more time would be beneficial because more schools would be able to participate. By the start of the expedition, most teachers may have had only two periods—the first two of the school year—with the classes that they could propose the participation in the competition to. Some have not participated because there

seemed to be too little time, and one school submitted their entries too late, after the winners were already selected and announced. To make this clearer, we have now expanded these statements as follows:

> "Two weeks is a short time for a teacher who may meet their science or geography class only once a week. Having more time for the competition would have been beneficial as more classes would be likely to participate. However, this was not possible as the expedition started shortly after the beginning of the academic year."

Regarding "show and tell," this is a case study, and showing and telling the reader how the programme was developed is an essential part of the paper. The evaluation aspects of the study are, of course, also essential. Following the reviewers' suggestions, we have substantially improved the presentation of the evaluation aspects in the revised version of the manuscript. Evaluation-survey data are now included in the Supplement.

The reviewer's "primarily anecdotal" suggestion is a gross exaggeration. The piece of our text given as an example has nothing anecdotal about it: it is an explanation of the general problem presented by the start of an expedition close to the beginning of the school year. This argument has now been expanded and clarified further.

*b. Lelliott indicates that the authors tend to make claims unsupported by any data – see e.g. p.6 line 194 – the assertion is that the girls in all-girl schools identify with female scientists as role models; this assertion is extended to include boys in the abstract (page 1 line 25, "both girls and boys…were presented with engaging role models"), and although teachers attested to the positive response of their pupils, the statements are anecdotal rather than substantiated with provable data.*

We saw this with our own eyes, the girls discovering that they could have a fun and informative discussion with young female scientists and developing a connection with them. We do not have "provable" data for this, however. We thus rephrase the sentence as follows:

> "This offered opportunities to all-girl classes and girls in co-educational schools to connect to and identify with their own role models among the scientists."

The revised statement is weaker but certainly accurate.

*In line 199 also page 6, the authors refer to broadened "impact" of the event through pupils talking to their friends and parents about the experience – this certainly can be said to broaden the "reach" of the event, but should not be assumed to be indicative of impact unless that impact can be established and substantiated.*

"impact" replaced with "reach"

*Further, at line 225, there is reference to the competition as having "increased the student's interest in STEM" – again there is no data to prove this.*

After this sentence in the text, we have added a reference to Section 4 (Evaluation), where this is explained. In Section 4 (Evaluation) itself, we now explain in greater detail how these inferences were made. The new text on this reads:

> "The evaluation relating to the drawing competition was informal, based on the feedback from the primary school teachers and students themselves. The bulk of the feedback came in the form

of 24 thank-you cards from 7-8 year old students from Abbeyleix South National School, Co. Laois, Ireland. It was clear that the children were encouraged to write the cards by their teacher and that the teacher must have mentioned a number of things that could be included in the cards. However, different children opted to include different things in their text, and the phrasing was their own, different in different cards. Nearly all the students wrote that they enjoyed learning about the seismometers and the project, with some mentioning explicitly that they explained what they had learned to their parents. All were pleased with their prizes, and happy that everybody got a prize. Some wrote that they would like to work with the SEA-SEIS researchers in the future.

This evidence would not yield robust statistical inferences (only one class, influenced by their teacher) but, even though the evidence may be regarded as anecdotal, we consider it encouraging and useful. It confirms that young primary school students are curious about and receptive to the general ideas of Earth science research, and that rewarding every participant with a prize is an effective approach in primary school competitions. It also highlights the key role of an actively participating teacher and the importance of a teacher network in order for such competition to reach a broad, national scale."

*c. An example where a conclusion can be linked to the project under discussion to avoid it reading as an unsubstantiated assumption, can be found on page 10, line 332: adding the words "We found that" to the sentence, "Inviting students to become co-creators gets them engaged with enthusiasm" and providing some examples to illustrate the claim, would substantiate the statement which otherwise should be referenced to be considered valid.*

"We found that" added.

*Further, how do the authors gauge that "They [the students] are, then, motivated to learn more…" or that "This contributes to increasing the students' interest in ST and STEM careers" (lines 338-39)?*
The sentence has been rephrased in a weaker form:

"This, in turn, is likely to contribute to increasing the students' interest in STEM and STEM careers."

*4. Are the results sufficient to support the interpretations and conclusions?*
*a. Reference to sample size (numbers), where it is made, is imprecise – e.g. Reference is made in the abstract to "18 link-ups", but in line 126 the authors state that "Around 20 schools… participated" – is this 18 -19 schools, 21 schools? The authors need to be consistent in stating the size of the sample set. Should it be the intention to use this paper as a benchmarking exercise for future engagements of this sort, it will be difficult to interrogate data and draw meaningful comparisons. I suggest they indicate exactly how many schools, how many pupils and how many teachers participated – they have provided a distribution map of the schools involved in the project which is useful, and obviously worked closely with teachers at the schools, so substantiated data should be really available.*

These numbers are different because they refer to different components of the EPE programme. There were 18 live video link-ups from the ship to classrooms. Prior to that, there was a seismometer naming

competition, to which 20 schools sent entries. These different activities are covered clearly in the text, each in its own subsection.

More generally, we agree that precise numbers are preferable and have replaced "~20 schools" with "20 schools" (having verified the number). We have also included the results of our evaluation survey in the supplement. This gives the numbers of students participating in different video links.

*b. Page 8, lines 251-254: In interpreting the results of the competitions, the authors refer to "many" students having "made an effort to research the subject" – again finite number should be available – this could provide useful comparative data for later competitions. On page 11, line 345-347, "the number of participating schools was lower than expected ", and "a proportion of entries showed little evidence of the students having researched…" – But, how do they confirm this is the case and if so what were the factors behind this? Investigating these questions could provide very valuable insights for geoscience education and communication going forward.*

To clarify this, the following text now concludes the subsection 3.4:

> "At one end of the spectrum, one of the winning compositions referred to most SEA-SEIS seismometers by their names, also with a reference to their location and to what they were recording on the seafloor, which showed substantial research on the subject. At the other end, some of the entries showed little understanding and probably no research behind them.
>
> Drawing a line separating "research" from "no research," however, would be difficult. The competition was developed in order to combine art and science, to get the students create science-themed art and to get them more interested in science. The pieces the students composed and recorded were free-form, and there were no correct answers they could insert into in their songs. For this reason, it is not possible to gauge precisely the amount of research students put into Earth-science research in the course of this activity.
>
> By the very nature of the art-science approach, it is not always possible to measure everything. We do draw a lesson from the first edition of this competition, however: in its future editions, it will be useful to steer students more firmly towards learning and towards communicating science and technology in their pieces."

More generally, we very much agree that more work on mixing art and science will be valuable for geoscience education and communication.  But, by the very nature of this approach, it is not always possible to measure everything precisely.

*c. In 'Conclusions', Page 11, lines 367-68, the authors affirm that "Survey responses from the teachers confirmed that the engagement…has a lasting positive impact": I would contend that a single survey with teachers only cannot be said to do this – a claim for "lasting…impact" needs to be measured over time through at least one follow-up survey conducted sometime after the engagement; firstly with pupils (to ascertain retention of knowledge and impact on their view/ understanding of geosciences) and teachers – with regard to how the engagement has supported or augmented their curriculum-based work in the classroom and influenced (especially in the case of high-school pupils) their performance and results.*

"Lasting impact" was used correctly in the context of the sentence: whereas the enjoyment of the activity is limited to its duration, its effect (encouragement of the students' interest in STEM) is more lasting. We can see, however, how this can get misleading if interpreted differently. We thus remove "lasting impact" and rephrase the sentence in the Conclusions as follows:

> "Survey responses from the teachers confirmed that the video links encouraged the students' interest in STEM."

The encouragement is evidenced by the evaluation-survey responses included in the Supplement.

*As far as the "researchers – both experienced and early career", there are a number of considerations that could be used to generate conclusions regarding the success of the outreach:*

*i. E.g. On page 9, line 299, researchers' communications skills are described as, "often not their greatest strength, to begin with": As individuals, how confident were the researchers on this project about being involved in geosciences engagement and communication before the project? Had this changed afterwards? Why? What are some of the take-home messages that other researchers undertaking engagement might find useful?*

The main take-home message relating to researchers in EPE, which we emphasize very clearly in the paper, is that "EPE coupled with active research projects can channel the academic researchers' drive and creativity into the development of spectacular, novel EPE programmes" (Section 5.1). This does not take anything away from other EPE approaches, or from (undoubtedly useful) communication training, which may come to mind here, but highlights the potential of EPE programmes coupled with active research projects, which is the focus of this paper.

Regarding the change in attitude and confidence of the particular researchers involved in this project (that is, a few of this paper's co-authors), we think that the dataset emerging from a survey within our group would probably lack statistical significance, because of its small size and the probable biases due to the participants' enthusiasm for the project. A study across a number of projects, conducted by an independent EPE researcher, would be more suitable to identify useful patterns and analyse them, but this is beyond the scope of this paper.

*ii. What were their expectations before the engagement? Did they feel these had been met? Where were the gaps? The authors answer this to some extent by mentioning that the programme should offer more activities (line 356) and broaden the target audience (lines 358-60) in order to achieve their aim (stated in line 353), but it would have been interesting to know how they are going about this – they mention that funding for expansion is "—being sought at the moment" (line 361), so they must feel that the programme was sufficiently successful to demonstrate the need. Who are the targeted funders? Motivating the need for enhanced and broader STEM awareness could be pitched to a variety of potential funders (government, the private sector, industry) but would need to address not only the educational but also societal and economic benefits. How does "the real impact" of the programme move beyond generating "enthusiasm" among researchers and address broader issues to which they can commit their energies and work with societal actors in order to provide answers?*

We agree that the need for enhanced and broader STEM awareness could be pitched to a variety of potential funders. And yes, the programme was successful and demonstrated the potential of the approach. In order to grow, expand and produce greater impacts, it needs more resources. We have written a proposal for funding that could support this, now submitted to the Discover Programme of

Science Foundation Ireland. There are various ideas in there that we are fond of, but at this stage the proposal is still under review and the ideas have not yet been implemented, so we should probably avoid talking too much of future plans and focus instead on what has been done in the present programme.

We have now included a new, second-to-last sentence of Section 5.5 to outline our general ideas for the expanded programme:

> "Using the approaches, lessons and recipes from the present programme, this can be addressed through the work with schools supported by the development of an effective teacher network, through presenting science through arts, and through the use of a pervasive digital platform."

Regarding specific details on a pending grant application (funder, programme, etc.), we probably should not go into them in the paper.

*5. Do the authors give proper credit to related work and clearly indicate their own new/original contribution?*

*a. At line 159, the authors indicate they were "keen to use the available best practice" which is acknowledged and credited. However, the reader is not able to compare their approach with other examples as those examples aren't articulated.*

Following the suggestion, we have now modified this and following sentences:

> "Following IODP, we started the video link-ups with an introduction from the ship that set the stage for a question and answer session with the students in the classroom. In contrast to the IODP video events, which, with rare exceptions, are led by educators (Kulhanek et al., 2014), all our video links were led by PhD students onboard.  We also had to develop our own template for live video events. Because the weather during our expedition was mostly stormy—which is often the case in the North Atlantic,—the initially planned live tours of the ship, as often performed on RV JOIDES Resolution's scientific cruises (Kulhanek et al., 2014), would have been dangerous for the presenters and had to be abandoned."

*Later, on p.12 line 382, they say that "No primary data sets were used in producing this article". Some of the best practice examples may have provided a potential source for comparative data and there are enough statements of "fact", or implied measures of comparison, in the article that could be supported by data.*

We have now provided the evaluation data in the Supplement.

*b. Not having explained what, in their view, constitutes best practice earlier in the article (ref line 106-7), it is not clear why the project leaders decided to develop a template of their own – was this because there were no examples/reports of live video events available from the best practice examples they had chosen? This would provide an opportunity for them to discuss the implied gaps in the literature and how and why their approach is new, different and an innovative departure from established geoscience communication practice.*

We have now expanded substantially the overview of the state of the art in the beginning of Section 3 (following the lines mentioned by the reviewer), as detailed above. We also give an example of where we had planned to follow the templates of RV JOIDES Resolution's scientific cruises but had to depart

from them and develop or own. To make this clearer, we expand the sentence in the second paragraph of Section 3.2 as follows:

> "Because the weather during our expedition was mostly stormy—which is often the case in the North Atlantic,—the initially planned live tours of the ship, as often performed on RV JOIDES Resolution's scientific cruises (Kulhanek et al., 2014), would have been dangerous for the presenters and had to be abandoned."

*6. Does the title clearly reflect the contents of the paper?*

*a. To an extent, but I'm not convinced that the content has fully articulated the "lessons and recipes" referred to in the title.*

For clarity, lessons are recipes are now mentioned and discussed explicitly and specifically throughout the text (Introduction, Discussion, Conclusions). The added new text is as follows.

End of the Introduction:

> "We describe the best practice developed in the course of the programme, draw lessons from its development, discuss some general inferences, and aim to identify useful templates and recipes for EPE projects that connect researchers to school students and to the general public."

Beginning of the Discussion:

> "In this section, we focus on the lessons and recipes provided by our EPE programme."

End of Conclusions:

> "This case study offers useful lessons and recipes for EPE programmes coupled with active research projects. First of all, it highlights how the research projects and the researchers working on them are a rich resource for EPE. Researchers can be effective EPE leaders, with the EPE programmes channelling their drive and creativity into the development of effective, novel EPE activities.
>
> Secondly, it illustrates the importance of an EPE team with diverse backgrounds and complementary expertise and skill sets. In EPE with primary and secondary schools, the most essential partners are the school teachers and principals. The development of a network of actively engaged teachers is a pre-requisite of successful EPE with schools and should, if possible, be initiated at the earliest stages of the programme. Beyond that, our programme capitalized on contributions of not only scientists and teachers, but also engineers and technicians and a sound artist. Effective national media campaign around the start of the SEA-SEIS Expedition and EPE programme was orchestrated by our communication manager, working with a partner PR firm. Our digital media presence was maintained primarily by the team's digital journalist, his expertise increasing its effectiveness greatly.
>
> Thirdly, the programme demonstrates the value of co-creation by the EPE team, teachers and school students. Close collaboration with the teachers was essential in planning and developing the programme activities. Getting the school students co-creating with scientists in the course of the school competitions got them engaged and genuinely interested.
>
> Finally, our project can be seen as a template for a multi-faceted, "low-cost, high-gain" EPE programme. Recognising that the school curricular are already packed, so that teachers find it difficult to allocate a lot of time to extra material, we offered them the flexibility of activities that could be fit into just one or two periods or integrated into the Earth science part of the

curriculum, according to their needs at the time. We offered a series of different, inter-related but independent activities (the naming competition prior to the expedition, live video events during the expedition, and the song and drawing competitions after the expedition). Some classes participated in only one of the activities, others—in two or three, with the teachers choosing what was the most suitable for them. In the sense of the minimum required classroom time commitment, the activities were low-cost. The high gain, relative to the amount of class-time investment and in terms of increasing the students' interest in STEM and STEM careers, is achieved thanks to the captivating adventure aspect of the project's fieldwork, engagement of students through co-creation with scientists, and direct, live-video communications between students and scientists."

Also, in the end of Abstract, we add new text that does not mention the words lessons and recipes (to avoid their excessive repetition) but summarizes briefly the main areas where the lessons and recipes are offered by the programme:

"The programme shows how research projects and the researchers working on them are a rich resource for EPE, highlights the importance of an EPE team with diverse backgrounds and expertise, and demonstrates the value of co-creation by the EPE team, teachers and school students. It also provides a template for a multi-faceted EPE programme that school teachers can use with flexibility, without extra strain on their teaching schedules."

*b. I agree with Tony Lelliott that the title is too long: Suggestion: "Ship to shore – live video involves Irish schools in an active geophysical research project in the North Atlantic."*

We appreciate the suggestion. We did put a lot of thought into our original title, however, and would very much prefer to keep it. The title suggested by Reviewer 2 would limit the stated scope of the programme to only one of its elements, the live video links, which would not be accurate, given that three school competitions before and after the video links have broadened the scope of the programme substantially and made it substantially more original. A part of our motivation for writing this paper came after our talk on the programme in an EPE session at the last EGU meeting, when a colleague working in a space research centre asked if we had a text on our programme that we could share, so that she could use some of our approaches in her outreach work. The first part of the title thus has an appropriately broad scope. As this is a case study, it is also important to spell out what this programme was, specifically. If the title was just "Education and public engagement using an active research project" or similar, it would give the wrong impression that this is a review covering the very broad area in some detail. Finally, the "lessons and recipes" are appropriate to mention as this is where most of the general usefulness of the paper probably is. For clarity, lessons are recipes are now mentioned and discussed explicitly throughout the text (Introduction, Discussion, Conclusions).

The title is, indeed, relatively long, but not exceptionally so: a number of titles on the Geoscience Communication Most Downloaded page (https://www.geosci-commun.net/most_downloaded.html), for example, are similar in length or longer.

*7. Does the abstract provide a concise and complete summary?*

*a. Yes, but it should be edited to address some of the comments above: e.g. see ref to "boys and girls" being "presented with engaging role models"*

We have, indeed, presented the school students with engaging, female and male role models – the young, enthusiastic PhD students with whom the school students often had exciting discussions. We have only anecdotal evidence for the effect that this has had, and for this reason we do not talk of this. But the fact that role models were presented is not in doubt.

*b. I suggest some of the chosen adjectives in the abstract should be changed to avoid implied assumptions or bias: e.g.*

*i. 'Profound' impact (line 17) – the content doesn't show that the impact of the project has been 'profound'.*

The sentence neither states nor implies that our project had a profound impact. This introductory sentence emphasizes the importance of the goals we are setting for ourselves: "Engagement with schools, in particular, can have a profound impact, showing the students how science works in practice, encouraging them to study science, and broadening their career perspectives."

*ii. Line 20-21 – the relative pronoun 'them' in "…which got 'them' enthusiastically engaged" is unclear – who became enthusiastically engaged – the students or the researchers?*

"them" has been replaced with "the students"

*8. Is the overall presentation well-structured and clear?*

*Yes.*

*9. Is the language fluent and precise?*

*a. On the whole the article is well-written.*

*b. Some of the expression/idiom is unfamiliar to me – e.g. line 133 "curriculum-facing discussions" – are these curriculum-'based' or curriculum-'relevant'? If the discussions are 'relevant' to the curriculum they could require some interpretation, critical thinking and application by the students; whereas curriculum-'based' discussion would be more guided and possibly require less application and critical thinking. What did the students learn from these discussions?*

"curriculum-facing" is, indeed, not commonly used. What we meant was "curriculum-aligned." Upon further reflection, however, we would like to simply remove the "curriculum-facing." We know that some of the teachers were teaching plate tectonics and volcanoes at the time of the competition, and discussing geoscience and geoscientists for the competition fit in nicely with the curriculum for them. But other teachers were not, so that the degree of curriculum alignment varied and a general statement on this would have to be weak and thus not very useful. The sentence now reads:

> "The teachers used this to have discussions on geoscience and marine science, as well as Irish and international Earth scientists and explorers."

*c. Tone: there are two instances where the 'tone' could be interpreted as indicating a 'bias' on the part of the authors:*

*i. On p.4 line 119, the use of the word "imaginative" in the context denotes a certain implied irony – whilst the intention may be to instil some humour, it falls flat. On the other hand it could be interpreted as a typographical error for "unimaginative".*

This was self-irony, not bias. However, seeing that this phrasing can be distracting rather than helpful in this paper, we have now removed the word "imaginative" and modified the sentence as follows:

> "The seismologists among us have a natural tendency to give their seismic stations names like S01, S02, S03, etc."

*ii. On p. 5 line 136-37, the parenthetical reference to the deployment of "Charles" and "Harry", "inevitably, in the UK waters", whilst attempting humour, could be seen as inappropriate in a scholarly work. It assumes also that a global readership would understand and accept the 'joke'.*

"(deployed, inevitably, in the UK waters)" has been removed. The sentence now reads:

> "…, and Charles and Harry–after the American seismologists Charles Richter and Harry Hess."

*d. Minor errors:*

*i. Line 370 – insert the word "can" after "research project" for the sentence from line 369 to read "The outcomes of an educational programme coupled with a research project can include…"*

"can" inserted, modifying the sentence.

*ii. Line 378 – "form" should read "from".*

Typo fixed – thank you.

*10. Are the number and quality of references appropriate?*

*Yes. However, the formatting of the reference list is not reader-friendly and makes the references difficult to tell apart. I suggest that the second and following lines of the references are indented to set the references apart.*

We have used the Geoscience Communication manuscript template exactly as it is.

*11. Additional comments:*

*a. I couldn't find any reference to ethical clearance for the research as required by the scope of GC. As the authors are gathering data from schools, and teachers and pupils are case subjects, the research proposal should have received ethical clearance. Is this an oversight?*

A new Ethics subsection has been added:

> "3.5 Ethics.
>
> The study complied with the Guidance for developing ethical research projects involving children (Department of Children and Youth Affairs, 2012). No personal information on children was collected. No interactions of project participants with children in the participating schools took place, other than the live video link-ups between the researchers and the classrooms, which were conducted by the teachers on the classroom side. The photographs of the children were supplied by the teachers, who confirmed the consent for their use in the online publication. Data collected in the evaluation survey of teachers were undertaken in accordance with good practice. The survey was anonymous by default. Contributors to this study were under no obligation to become the paper's co-authors."

*b. Protection of minors:*

*i. Is it acceptable to use photos of under age children on an online platform?*

Schools in Ireland and Italy obtain consent from parents for the use of their children's images, which is usually in the schools' promotional materials and social media. The photos used in our paper and similar ones have been used by the schools in their Tweeter feeds, for example. The teachers have sent the photos to us so that we could publish them. The teachers have shared only the photos for which they have parental consent, which we have confirmed with them.

[revised manuscript text omitted]

**Scientists on the ship**

**Students in class**

RV Celtic Explorer, North Atlantic

Lycée Français d'Irlande, Primary School, Dublin

**Figure 2:** Live, ship-to-classroom video link-ups started with a brief introduction of the project and the team and continued with a 20-70 minute Q&A session.

645

650

[Figure]

Scoil Mhuire, Buncrana, Co. Donegal

Gaelcholaiste Carrigaline, Co. Cork

Coláiste Phobail Cholmcille
Tory Island, Co. Donegal

Kingswood Community College, Dublin

Lamezia Terme,
Italy

**Figure 3:** Live, ship-to-classroom video link-ups with different schools in Ireland and Italy.

655

[Figure]

**Figure 4:** Participants of the Primary School Drawing Competition with their prizes, calendars featuring their art. Top left: the 2019-2020 calendar. Top right: students at Istituto Comprensivo Don Lorenzo Milani, Lamezia Terme, Italy. Bottom, left and right: students at Abbeyleix South National School, Abbeyleix, Co. Laois, Ireland.

660

[Figure]

[Figure]

[Figure]

**Figure 5:** Prizes and some of the winners of the geoscience song-and-rap competition for secondary schools. Left:
665   Inspirational science books went to classes with winning and runner-up groups. Centre: one of the two Grand Prize winning
groups (Lycée Français d'Irlande, Dublin). Right: SEA-SEIS branded, 16GB flash drives were awarded to every student in
the winning groups and to their teachers.

*Supplement of*

**Education and public engagement using an active research project: lessons and recipes from the SEA-SEIS North Atlantic Expedition's programme for Irish schools**

**Sergei Lebedev et al.**

*Correspondence to*: Sergei Lebedev (sergei@cp.dias.ie)

This file presents the data yielded by the evaluation survey of participating teachers in Ireland following our live, ship-to-classroom video link-ups. The survey was conducted via SurveyMonkey (surveymonkey.com).

The survey included 10 questions. The first 5 were multiple-choice and the last 5 were free-form questions. In the following, we first give a graphical summary and basic statistics for the 5 multiple-choice questions. After that, we give complete responses by the respondents to all the questions, apart from the last one.

The survey was anonymous by default. The last, 10th question gave the participants an opportunity to give their name and information on their school and class, if they wished. Here, we omit the answers to this last question.

**Q1 How many stars (1 is worst, 5 is best), would you give your SEA-SEIS video link overall?**

Answered: 14    Skipped: 0

[Figure]

| | 1 (1) | 2 (2) | 3 (3) | 4 (4) | 5 (5) | TOTAL | WEIGHTED AVERAGE |
|---|---|---|---|---|---|---|---|
| ☆ | 0.00%
0 | 0.00%
0 | 7.14%
1 | 14.29%
2 | 78.57%
11 | 14 | 4.71 |

| BASIC STATISTICS | | | | |
|---|---|---|---|---|
| Minimum
3.00 | Maximum
5.00 | Median
5.00 | Mean
4.71 | Standard Deviation
0.59 |

**Q2 How was the audio and video quality?**

Answered: 14    Skipped: 0

[Figure]

| ANSWER CHOICES | RESPONSES | |
|---|---|---|
| Very poor (1) | 0.00% | 0 |
| Poor (2) | 0.00% | 0 |
| Satisfactory (3) | 14.29% | 2 |
| Good (4) | 64.29% | 9 |
| Excellent (5) | 21.43% | 3 |
| TOTAL | | 14 |

| BASIC STATISTICS | | | | |
|---|---|---|---|---|
| Minimum 3.00 | Maximum 5.00 | Median 4.00 | Mean 4.07 | Standard Deviation 0.59 |

**Q3 How well did the students understand why the scientists went on the expedition?**

Answered: 14    Skipped: 0

[Figure]

| ANSWER CHOICES | RESPONSES | |
|---|---|---|
| Not at all (1) | 0.00% | 0 |
| A little (2) | 0.00% | 0 |
| Partially (3) | 7.14% | 1 |
| Reasonably well (4) | 57.14% | 8 |
| Very well (5) | 35.71% | 5 |
| TOTAL | | 14 |

| BASIC STATISTICS | | | | |
|---|---|---|---|---|
| Minimum 3.00 | Maximum 5.00 | Median 4.00 | Mean 4.29 | Standard Deviation 0.59 |

**Q4 Was the information presented on the appropriate level for your audience?**

Answered: 14    Skipped: 0

[Figure]

| ANSWER CHOICES | RESPONSES | |
| --- | --- | --- |
| No (1) | 0.00% | 0 |
| Some of it (2) | 7.14% | 1 |
| Yes (3) | 92.86% | 13 |
| TOTAL | | 14 |

| BASIC STATISTICS | | | | |
| --- | --- | --- | --- | --- |
| Minimum | Maximum | Median | Mean | Standard Deviation |
| 2.00 | 3.00 | 3.00 | 2.93 | 0.26 |

**Q5 Did the video link encourage the students' interest in science?**

Answered: 14    Skipped: 0

[Figure]

| ANSWER CHOICES | RESPONSES | |
|---|---|---|
| No (1) | 0.00% | 0 |
| Somewhat (2) | 14.29% | 2 |
| Yes (3) | 85.71% | 12 |
| TOTAL | | 14 |

| BASIC STATISTICS | | | | |
|---|---|---|---|---|
| Minimum 2.00 | Maximum 3.00 | Median 3.00 | Mean 2.86 | Standard Deviation 0.35 |

**#1**

**COMPLETE**

| | |
|---|---|
| **Collector:** | Web Link 1 (Web Link) |
| **Started:** | Wednesday, September 26, 2018 9:29:06 PM |
| **Last Modified:** | Wednesday, September 26, 2018 9:33:57 PM |
| **Time Spent:** | 00 : 04 : 50 |
| **IP Address:** | |

**Q1** How many stars (1 is worst, 5 is best), would you give your SEA-SEIS video link overall?

☆                                                  **5**

**Q2** How was the audio and video quality?      **Excellent**

**Q3** How well did the students understand why the scientists went on the expedition?      **Very well**

**Q4** Has the information presented on the appropriate level for your audience?      **Yes**

**Q5** Did the video link encourage the students' interest in science?      **Yes**

**Q6** Did the students enjoy the video link? Did it spark their imagination?

They were telling their other teachers about it afterwards so it was definitely enjoyed. They were asking could they keep an eye on how you were getting on in class !

**Q7** What did you like most about your video link?

The crew were very good with the students and obviously loved their jobs which came across very well

**Q8** Approximately how many people were in attendance at your event?

16 plus teacher

**Q9** Do you have any general comments or suggestions for our video links?

Keep up the good work !!

**#2**

**COMPLETE**

| | |
|---|---|
| **Collector:** | Web Link 1 (Web Link) |
| **Started:** | Wednesday, September 26, 2018 9:37:37 PM |
| **Last Modified:** | Wednesday, September 26, 2018 9:41:19 PM |
| **Time Spent:** | 00:03:42 |
| **IP Address:** | |

**Q1** How many stars (1 is worst, 5 is best), would you give your SEA-SEIS video link overall?

☆                                                                     **5**

**Q2** How was the audio and video quality?                           **Good**

**Q3** How well did the students understand why the                  **Reasonably well**
scientists went on the expedition?

**Q4** Has the information presented on the appropriate               **Yes**
level for your audience?

**Q5** Did the video link encourage the students' interest            **Yes**
in science?

**Q6** Did the students enjoy the video link? Did it spark their imagination?

They enjoyed it and continue to talk about it. Some wants to participate to the competition!

**Q7** What did you like most about your video link?

the questions with the team!

**Q8** Approximately how many people were in attendance at your event?

**Q9** Do you have any general comments or suggestions for our video links?

no, it was nice

**#3**

**COMPLETE**

| | |
|---|---|
| **Collector:** | Web Link 1 (Web Link) |
| **Started:** | Wednesday, September 26, 2018 9:47:50 PM |
| **Last Modified:** | Wednesday, September 26, 2018 9:50:47 PM |
| **Time Spent:** | 00:02:56 |
| **IP Address:** | |

**Q1** How many stars (1 is worst, 5 is best), would you give your SEA-SEIS video link overall?

☆                                                                    **5**

**Q2** How was the audio and video quality?                          **Excellent**

**Q3** How well did the students understand why the scientists went on the expedition?                          **Very well**

**Q4** Has the information presented on the appropriate level for your audience?                          **Yes**

**Q5** Did the video link encourage the students' interest in science?                          **Yes**

**Q6** Did the students enjoy the video link? Did it spark their imagination?

Yes, they loved it. They enjoyed seeing scientists from different strands and backgrounds working on the same project.

**Q7** What did you like most about your video link?

Great atmosphere from scientists, very friendly and engaging towards students.

**Q8** Approximately how many people were in attendance at your event?

**Q9** Do you have any general comments or suggestions for our video links?

Excellent idea. Really sparked and interest in the students

**#4**

**COMPLETE**

| | |
|---|---|
| **Collector:** | Web Link 1 (Web Link) |
| **Started:** | Wednesday, September 26, 2018 9:44:05 PM |
| **Last Modified:** | Wednesday, September 26, 2018 9:51:55 PM |
| **Time Spent:** | 00:07:49 |
| **IP Address:** | |
* * *
**Q1** How many stars (1 is worst, 5 is best), would you give your SEA-SEIS video link overall?

☆                                                             **4**
* * *
**Q2** How was the audio and video quality?             **Satisfactory**
* * *
**Q3** How well did the students understand why the scientists went on the expedition?          **Reasonably well**
* * *
**Q4** Has the information presented on the appropriate level for your audience?          **Yes**
* * *
**Q5** Did the video link encourage the students' interest in science?          **Yes**
* * *
**Q6** Did the students enjoy the video link? Did it spark their imagination?

Yes they really enjoyed it and other classes were asking them all about it - think they felt very privileged that it was just them getting this special lesson
* * *
**Q7** What did you like most about your video link?

The very friendly nature - smiling faces of all the scientists who spoke to and answered the students questions - this allowed usually very shy students to come forward to the camera.
* * *
**Q8** Approximately how many people were in attendance at your event?
* * *
**Q9** Do you have any general comments or suggestions for our video links?

The audio was difficult to hear for the pre recorded parts of scientists explaining their roles but got much better for questions and answers inside
* * *
**#5**

COMPLETE

| | |
|---|---|
| **Collector:** | Web Link 1 (Web Link) |
| **Started:** | Wednesday, September 26, 2018 10:06:12 PM |
| **Last Modified:** | Wednesday, September 26, 2018 10:11:59 PM |
| **Time Spent:** | 0 0 : 0 5 : 4 6 |
| **IP Address:** | |

**Q1** How many stars (1 is worst, 5 is best), would you give your SEA-SEIS video link overall?

☆ **5**

**Q2** How was the audio and video quality? **Good**

**Q3** How well did the students understand why the scientists went on the expedition? **Very well**

**Q4** Has the information presented on the appropriate level for your audience? **Yes**

**Q5** Did the video link encourage the students' interest in science? **Yes**

**Q6** Did the students enjoy the video link? Did it spark their imagination?

yes! it showed them that science is part of real life and they are very pleased that the project is so close to their homes. One student is preparing work for another project and has decided to concentrate on Plate tectonics!

**Q7** What did you like most about your video link?

The fact that the 'scientists' who are normally perceived as a remote group of people removed from ordinary society were willing to talk to our students.
And the fact that the team seemed very happy to be on the ship and excited by the project

**Q8** Approximately how many people were in attendance at your event?

5 students (total pupil population); principal and three other teachers and the school administrator.

**Q9** Do you have any general comments or suggestions for our video links?

try and do more of them.
I appreciate the difficulties involved but it will encourage students when they see science in action.

**#6**

**COMPLETE**

| | |
|---|---|
| **Collector:** | Web Link 1 (Web Link) |
| **Started:** | Thursday, September 27, 2018 4:42:56 PM |
| **Last Modified:** | Thursday, September 27, 2018 4:45:47 PM |
| **Time Spent:** | 0 0 : 0 2 : 5 0 |
| **IP Address:** | |
* * *
**Q1** How many stars (1 is worst, 5 is best), would you give your SEA-SEIS video link overall?

☆                                                                    5
* * *
**Q2** How was the audio and video quality?            **Excellent**
* * *
**Q3** How well did the students understand why the scientists went on the expedition?            **Reasonably well**
* * *
**Q4** Has the information presented on the appropriate level for your audience?            **Yes**
* * *
**Q5** Did the video link encourage the students' interest in science?            **Yes**
* * *
**Q6** Did the students enjoy the video link? Did it spark their imagination?

Yes! The students very much enjoyed the video link! They can't believe how lucky they were to experience that and to see the seismometer being deployed!
* * *
**Q7** What did you like most about your video link?

The whole experience was fantastic! From the excitement waiting for the call to the questions and answers it was a great morning!
* * *
**Q8** Approximately how many people were in attendance at your event?
* * *
**Q9** Do you have any general comments or suggestions for our video links?

Keep up the good work!
* * *
**Q10** Optional: This survey is anonymous, but if you do not mind sharing this information, please give your name, the name of the school, and the class that participated.

Jessica Lynch. Kingswood Community College. 2nd & 3rd Year Geography classes.

**#7**

**COMPLETE**

| | |
|---|---|
| **Collector:** | Web Link 1 (Web Link) |
| **Started:** | Thursday, September 27, 2018 5:34:53 PM |
| **Last Modified:** | Thursday, September 27, 2018 5:40:25 PM |
| **Time Spent:** | 00:05:31 |
| **IP Address:** | |

**Q1** How many stars (1 is worst, 5 is best), would you give your SEA-SEIS video link overall?

☆  **5**

**Q2** How was the audio and video quality?  **Good**

**Q3** How well did the students understand why the scientists went on the expedition?  **Reasonably well**

**Q4** Has the information presented on the appropriate level for your audience?  **Yes**

**Q5** Did the video link encourage the students' interest in science?  **Somewhat**

**Q6** Did the students enjoy the video link? Did it spark their imagination?

The students were full of energy after the video and were still coming up with questions they should have asked at the end of the day. This showed that they were thinking about the information they learnt and about the answers to other peoples questions.

**Q7** What did you like most about your video link?

Getting to meet all the staff, and the variety of different expertises that are on the both. I think it showed the importance of collaborating, and the different sides there are to a real life functioning scientific team.

**Q8** Approximately how many people were in attendance at your event?

**Q9** Do you have any general comments or suggestions for our video links?

I only booked the room for single class 40min but I think the students would have kept going for longer had there been time to do so. I thought that would be sufficient time. Had I known it might run over I would have organised an extended period.

**#8**

**COMPLETE**

| | |
|---|---|
| **Collector:** | Web Link 1 (Web Link) |
| **Started:** | Thursday, September 27, 2018 5:54:17 PM |
| **Last Modified:** | Thursday, September 27, 2018 5:58:57 PM |
| **Time Spent:** | 00:04:40 |
| **IP Address:** | |

**Q1** How many stars (1 is worst, 5 is best), would you give your SEA-SEIS video link overall?

☆                                                          **5**

**Q2** How was the audio and video quality?                **Good**

**Q3** How well did the students understand why the scientists went on the expedition?                **Reasonably well**

**Q4** Has the information presented on the appropriate level for your audience?                **Yes**

**Q5** Did the video link encourage the students' interest in science?                **Yes**

**Q6** Did the students enjoy the video link? Did it spark their imagination?

I would say that they loved it.

**Q7** What did you like most about your video link?

I loved it all.
The crew introductions and information at the start was very well done. The question and answer session was very enjoyable and interesting. Your crew were very friendly.

**Q8** Approximately how many people were in attendance at your event?

**Q9** Do you have any general comments or suggestions for our video links?

It was very well planned - well done!
The slight technical problems were our end and didn't last long.
My only suggestion would be to show us a bit more of the ship.

**#9**

**COMPLETE**

| | |
|---|---|
| **Collector:** | Web Link 1 (Web Link) |
| **Started:** | Thursday, September 27, 2018 6:25:36 PM |
| **Last Modified:** | Thursday, September 27, 2018 6:29:49 PM |
| **Time Spent:** | 00:04:13 |
| **IP Address:** | |

**Q1** How many stars (1 is worst, 5 is best), would you give your SEA-SEIS video link overall?

☆ 	5

**Q2** How was the audio and video quality? 	**Good**

**Q3** How well did the students understand why the scientists went on the expedition? 	**Reasonably well**

**Q4** Has the information presented on the appropriate level for your audience? 	**Yes**

**Q5** Did the video link encourage the students' interest in science? 	**Yes**

**Q6** Did the students enjoy the video link? Did it spark their imagination?

Yes they enjoyed it very much. They were amazed that they were going to part of a research project collecting data that had never been collected before.

**Q7** What did you like most about your video link?

The question and answer session was excellent and really informative. That was the mine and the students highlight

**Q8** Approximately how many people were in attendance at your event?

**Q9** Do you have any general comments or suggestions for our video links?

No. Excellent quality and delivered in a very efficient and interesting manner.

**#10**

**COMPLETE**

| | |
|---|---|
| **Collector:** | Web Link 1 (Web Link) |
| **Started:** | Thursday, September 27, 2018 6:34:43 PM |
| **Last Modified:** | Thursday, September 27, 2018 6:41:56 PM |
| **Time Spent:** | 00:07:13 |
| **IP Address:** | |

**Q1** How many stars (1 is worst, 5 is best), would you give your SEA-SEIS video link overall?

☆          **4**

**Q2** How was the audio and video quality?      **Good**

**Q3** How well did the students understand why the scientists went on the expedition?      **Partially**

**Q4** Has the information presented on the appropriate level for your audience?      **Some of it**

**Q5** Did the video link encourage the students' interest in science?      **Somewhat**

**Q6** Did the students enjoy the video link? Did it spark their imagination?

It was very exciting for the class (1st year students aged 12/13) and tied in well with the the topic on plate tectonics which we had recently covered.
The students really enjoyed the experience and I have no doubt that it broadened many of their career ideas!

**Q7** What did you like most about your video link?

The fact that we were communicating with people far away in the middle of the ocean in real time!

**Q8** Approximately how many people were in attendance at your event?

**Q9** Do you have any general comments or suggestions for our video links?

No, it was all went very well.

**#11**

**COMPLETE**

| | |
|---|---|
| **Collector:** | Web Link 1 (Web Link) |
| **Started:** | Monday, October 01, 2018 3:54:59 PM |
| **Last Modified:** | Monday, October 01, 2018 4:00:41 PM |
| **Time Spent:** | 0 0 : 0 5 : 4 1 |
| **IP Address:** | |

**Q1** How many stars (1 is worst, 5 is best), would you give your SEA-SEIS video link overall?

☆ **5**

**Q2** How was the audio and video quality? **Good**

**Q3** How well did the students understand why the scientists went on the expedition? **Reasonably well**

**Q4** Has the information presented on the appropriate level for your audience? **Yes**

**Q5** Did the video link encourage the students' interest in science? **Yes**

**Q6** Did the students enjoy the video link? Did it spark their imagination?

Definitely yes, it triggered their curiosity greatly. They were asking more questions after the link.

**Q7** What did you like most about your video link?

Photos of the named seismometer. Asking questions. Idea of gaining information about tsunamis etc.

**Q8** Approximately how many people were in attendance at your event?

**Q9** Do you have any general comments or suggestions for our video links?

Maybe some video footage of where the scientists sleep, eat and work.

**#12**

COMPLETE

| | |
|---|---|
| **Collector:** | Web Link 1 (Web Link) |
| **Started:** | Tuesday, October 02, 2018 9:17:53 PM |
| **Last Modified:** | Tuesday, October 02, 2018 9:29:35 PM |
| **Time Spent:** | 00:11:42 |
| **IP Address:** | |

**Q1** How many stars (1 is worst, 5 is best), would you give your SEA-SEIS video link overall?

| ☆ | **5** |
|---|---|

| **Q2** How was the audio and video quality? | **Good** |
|---|---|

| **Q3** How well did the students understand why the scientists went on the expedition? | **Very well** |
|---|---|

| **Q4** Has the information presented on the appropriate level for your audience? | **Yes** |
|---|---|

| **Q5** Did the video link encourage the students' interest in science? | **Yes** |
|---|---|

**Q6** Did the students enjoy the video link? Did it spark their imagination?

Yes the students enjoyed the video link and they learned a lot through discussion with the Scientists .

**Q7** What did you like most about your video link?

The video where the seismometer was dipping into the water .

**Q8** Approximately how many people were in attendance at your event?

**Q9** Do you have any general comments or suggestions for our video links?

Everything was very good .

**#13**

COMPLETE

| | |
|---|---|
| **Collector:** | Web Link 1 (Web Link) |
| **Started:** | Wednesday, October 03, 2018 4:42:10 PM |
| **Last Modified:** | Wednesday, October 03, 2018 4:48:16 PM |
| **Time Spent:** | 0 0 : 0 6 : 0 6 |
| **IP Address:** | |

**Q1** How many stars (1 is worst, 5 is best), would you give your SEA-SEIS video link overall?

☆                                                               **5**

**Q2** How was the audio and video quality?                   **Good**

**Q3** How well did the students understand why the scientists went on the expedition?                    **Very well**

**Q4** Has the information presented on the appropriate level for your audience?                   **Yes**

**Q5** Did the video link encourage the students' interest in science?                   **Yes**

**Q6** Did the students enjoy the video link? Did it spark their imagination?

Yes, absolutely

**Q7** What did you like most about your video link?

The link with the real. The time spent by the team to answer our questions. Friendly but serious spirit

**Q8** Approximately how many people were in attendance at your event?

25 students and 3 adults

**Q9** Do you have any general comments or suggestions for our video links?

No, it was really interesting because it will introduce my lesson on volcanism.

**#14**

**COMPLETE**

| | |
|---|---|
| **Collector:** | Web Link 1 (Web Link) |
| **Started:** | Wednesday, October 03, 2018 5:28:07 PM |
| **Last Modified:** | Wednesday, October 03, 2018 5:42:46 PM |
| **Time Spent:** | 00:14:38 |
| **IP Address:** | |

**Q1** How many stars (1 is worst, 5 is best), would you give your SEA-SEIS video link overall?

☆ 3

**Q2** How was the audio and video quality?

**Satisfactory**

**Q3** How well did the students understand why the scientists went on the expedition?

**Reasonably well**

**Q4** Has the information presented on the appropriate level for your audience?

**Yes**

**Q5** Did the video link encourage the students' interest in science?

**Yes**

**Q6** Did the students enjoy the video link? Did it spark their imagination?

Those students made sure to go and tell others in their year about their experience and interests. They would have liked to be doing something like this every day.

**Q7** What did you like most about your video link?

The actual witnessing of the seismometer as it entered the waters off the North Atlantic Ocean.
The interaction between students and scientists and how open and friendly you were.

**Q8** Approximately how many people were in attendance at your event?

20 students

**Q9** Do you have any general comments or suggestions for our video links?

There was a bit of time lag in some of the recordings between what someone was saying and when it was said. It was confusing.